# Spectral Evolution Search: Efficient Inference-Time Scaling for Reward-Aligned Image Generation

Jinyan Ye [1]  Zhongjie Duan [2]  Zhiwen Li [1]  Cen Chen [1]  Daoyuan Chen [2]  Yaliang Li [2]  Yingda Chen [2]

## Abstract

Inference-time scaling offers a flexible way to align visual generative models with downstream objectives without updating model parameters. In modern image generation models, a natural way to do this is to optimize the random noise from which generation starts. However, searching in this high-dimensional noise space is highly inefficient, because many directions have little effect on the final image. We trace this inefficiency to a spectral bias in generative dynamics: model sensitivity to initial perturbations decays rapidly as frequency increases. Based on this insight, we propose Spectral Evolution Search (SES), a plug-and-play framework for initial noise optimization that performs gradient-free evolutionary search in a low-frequency subspace. Theoretically, we derive the Spectral Scaling Prediction from perturbation propagation dynamics, which explains the frequency-dependent impact of perturbations. Extensive experiments across diverse settings show that SES substantially improves the trade-off between generation quality and computational cost, consistently outperforming strong baselines under the same compute budget.

## 1. Introduction

Modern text-to-image models have achieved remarkable progress driven by training-time scaling laws (Kaplan et al., 2020; Podell et al., 2024; Esser et al., 2024; Wu et al., 2025). However, further aligning these foundation models with diverse downstream objectives, such as semantic adherence, aesthetic quality, or human preference, remains a challenge. Conventional approaches rely on fine-tuning (Black et al., 2024; Clark et al., 2024; Wallace et al., 2024; Liang et al.,

2025), which is inherently inefficient because it requires retraining a dedicated model for each specific objective and often incurs substantial computational cost. This limitation has drawn increasing research attention to *inference-time scaling* (Snell et al., 2024; Brown et al., 2024). Instead of altering model weights, this paradigm translates additional computational budget into alignment gains, enabling a static model to adapt to specific objectives during sampling.

Current inference-time scaling methods struggle to balance generality with efficient budget-to-alignment conversion in practice. Methods based on particle filtering, resampling, or path planning (Skreta et al., 2025; Dou & Song, 2024; Kim et al., 2025; Singhal et al., 2025; Li et al., 2025b; Zhang et al., 2025; Zhao et al., 2026) require intrusive intervention in the sampling process and are often tied to specific solvers, limiting compatibility with efficient ODE samplers and newer architectures such as flow matching. Gradient-based optimization (Wallace et al., 2023; Tang et al., 2025) instead requires differentiable objectives and is susceptible to reward hacking. These limitations make initial noise optimization (Ma et al., 2025; Chen et al., 2024; Guo et al., 2024) an appealing alternative, as it formulates inference as a black-box search over the starting noise. However, existing methods typically adopt an implicit isotropic parameterization that treats all search directions equally, wasting budget on perturbations with negligible effect on the final generation and reducing the overall efficiency of search.

To unlock the potential of initial noise optimization, we argue that search must operate on a compact subspace of meaningful degrees of freedom. As illustrated in Figure 1, qualitative analysis reveals a pronounced spectral bias in generative models: low-frequency perturbations significantly reshape image structure, whereas high-frequency perturbations of equivalent energy exert negligible visual impact. Consequently, we hypothesize that the effective control space for inference-time search is intrinsically sparse and concentrated within low-frequency components. Guided by this insight, we introduce *Spectral Evolution Search (SES)*, a plug-and-play framework designed for initial noise optimization. SES uses wavelet transforms to decouple the search space, strictly constraining optimization to the low-frequency subspace via a cross-entropy evolutionary strat-

---

[1] School of Data Science and Engineering, East China Normal University, Shanghai, China [2] Alibaba Group, Hangzhou, China. Correspondence to: Cen Chen <cenchen@dase.ecnu.edu.cn>.

*Proceedings of the 43rd International Conference on Machine Learning*, Seoul, South Korea. PMLR 306, 2026. Copyright 2026 by the author(s).

egy. To theoretically substantiate this spectral decoupling strategy, we derive the *Spectral Scaling Prediction* from the perturbation propagation dynamics of generative flows, formally characterizing the differential gains of perturbations across frequency bands. Experimental results demonstrate that, under the same computational budget, SES achieves superior scaling behavior, substantially improving the trade-off between generation quality and computational cost.

Our main contributions are summarized as follows:

- **Method.** We propose Spectral Evolution Search (SES), a gradient-free inference-time scaling framework for initial noise optimization. It is plug-and-play and broadly applicable across diverse models, samplers, and reward functions.

- **Theory.** We derive the *Spectral Scaling Prediction* from perturbation propagation dynamics, showing that the effective optimization landscape is strongly spectrally biased toward low frequencies.

- **Practice.** Extensive experiments across mainstream generative models and alignment tasks demonstrate that SES yields superior inference-time scaling behavior, significantly outperforming representative methods under identical compute budgets.

**Conflict of Interest Disclosure.** Authors Z.D., D.C., Y.L., and Y.C. are employed by Alibaba Group. This paper includes experiments using Qwen-Image (Wu et al., 2025), a model developed within Alibaba-related research efforts, as one of the base generative models. These authors were not involved in the development of Qwen-Image.

## 2. Related Work

**Inference-Time Scaling Strategies.** Inference-time scaling aims to push the performance boundaries of pre-trained models by allocating additional computational budget during inference to enhance generation quality. Existing strategies can be broadly divided into four categories: (1) *Particle filtering and resampling.* Methods such as Sequential Monte Carlo (SMC) (Skreta et al., 2025; Dou & Song, 2024; Kim et al., 2025; Singhal et al., 2025) and SVDD (Li et al., 2025b) correct the generative distribution by introducing particle resampling mechanisms at intermediate steps. (2) *Path planning.* These approaches formulate the generation process as a sequential decision-making problem, utilizing algorithms like Beam Search (Zhang et al., 2025; Li et al., 2025a) or Monte Carlo Tree Search (MCTS) (Zhao et al., 2026; Jain et al., 2026) to search for the optimal trajectory. However, these two categories of methods typically necessitate deep intervention within the denoising process and are coupled with specific solvers, making them difficult to

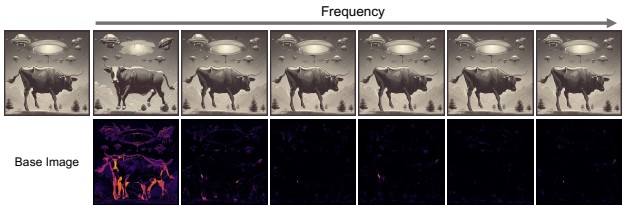

*Figure 1.* Visualization of spectral bias. We inject band-pass perturbations of constant energy ($\|\boldsymbol{\xi}\|_2 = 1$) into the initial noise across increasing frequency bands (left to right). The top row shows the generated samples, while the bottom row visualizes pixel-wise differences from the unperturbed baseline. Observe that low-frequency noise leads to significant changes, whereas high-frequency perturbations result in minimal visual difference.

directly adapt to novel architectures such as flow matching or efficient samplers. (3) *Gradient-based optimization.* Utilizing backpropagation to update noise (Wallace et al., 2023; Tang et al., 2025), these approaches are constrained to differentiable rewards and susceptible to reward hacking, exploiting metrics at the expense of perceptual fidelity. (4) *Initial noise optimization.* This paradigm (Ma et al., 2025; Chen et al., 2024; Guo et al., 2024) optimizes initial noise. While compatible with non-differentiable rewards, current approaches face the curse of dimensionality, leading to inefficient search under isotropic parameterizations of high-dimensional noise spaces.

**Spectral Bias in Diffusion Models.** Recent studies have begun to systematically analyze the imbalanced learning and generative behaviors of diffusion models in the frequency domain. Related works can be broadly categorized into two directions: (1) *Spectral bias analysis during training.* This line of research primarily investigates the mechanisms driving the "low-frequency first, high-frequency later" learning paradigm in diffusion models. It also explores how this phenomenon relates to noise scheduling, data spectral statistics, and model inductive biases. Perspectives from learning dynamics theory (Wang & Pehlevan, 2026), diffusion processes in Fourier space (Falck et al., 2025), and frequency-based noise control (Jiralerspong et al., 2025) indicate that low-frequency, high-variance modes are generally easier to learn. Furthermore, noise design significantly shapes the model's bias in the frequency domain. (2) *Frequency recovery and utilization during inference.* This direction focuses on the recovery dynamics of different frequency components during the reverse denoising process. Based on these insights, researchers have developed frequency-aware control and acceleration methods. For example, frequency-decoupled guidance (Sadat et al., 2025) can maintain higher sampling fidelity at low classifier-free guidance (CFG) scales. Additionally, frequency-aware caching mechanisms (Liu et al., 2025) leverage the imbalanced evolution characteristics across different frequency bands to reduce computational redundancy during inference.

## 3. Preliminaries

**Generative Flow as a Deterministic Mapping.** Modern visual generative models, including denoising diffusion (Ho et al., 2020; Rombach et al., 2022) and flow matching (Lipman et al., 2023; Esser et al., 2024), can be viewed under a unified continuous-time ODE framework. Despite differing training objectives, both paradigms essentially induce a deterministic trajectory that transports a prior noise distribution $p_0(\mathbf{x}) = \mathcal{N}(\mathbf{0}, \mathbf{I})$ to the data distribution $p_1(\mathbf{x}) = p_{\text{data}}(\mathbf{x})$. From this perspective, inference constitutes a deterministic mapping $\Psi_\theta(\cdot, c) : \mathcal{Z} \to \mathcal{X}$ from the Gaussian noise space $\mathcal{Z} = \mathbb{R}^d$ to the data space $\mathcal{X}$. Consequently, the generated sample $\mathbf{x}_1 = \Psi_\theta(\mathbf{x}_0, c)$ is uniquely determined by the initial noise $\mathbf{x}_0$, rendering it the primary controllable degree of freedom when the condition $c$ is fixed. However, the standard Gaussian prior $p_0(\mathbf{x})$ is inherently task-agnostic: its high-probability regions often fail to align with the noise regions yielding high-reward samples. This structural misalignment between the prior and target utility motivates inference-time initial noise optimization.

**Inference-Time Scaling as Black-Box Optimization.** We formulate inference-time alignment as a black-box optimization problem over the initial noise. Given a reward function $\mathcal{R}$, the objective is to locate the optimal initial noise $\mathbf{x}_0^*$ within the Gaussian noise space $\mathcal{Z}$:

$$\mathbf{x}_0^* = \arg\max_{\mathbf{x}_0 \in \mathcal{Z}} \mathcal{R}(\Psi_\theta(\mathbf{x}_0, c)). \tag{1}$$

From this perspective, inference shifts from a one-shot sampling event to an iterative optimization process centered on the initial noise. Following the inference-time scaling law (Snell et al., 2024; Brown et al., 2024), we characterize generation quality $\mathcal{Q}$ as a function of the inference budget $N$, measured by the *Number of Reward Evaluations (NRE)*. Under the evaluation protocol, each evaluated candidate produces one reward score, so NRE equals the number of evaluated candidates. We adopt this metric because scoring any candidate requires generating its final image, which consumes additional function evaluations. As a result, the conventional Number of Function Evaluations (NFE) conflates two factors: how many candidates are explored and how much computation is spent evaluating them. By contrast, NRE directly measures the reward feedback obtained during search, and thus more faithfully captures the sample efficiency of an inference-time scaling strategy. To ensure fair comparison, all rewards are computed on final samples produced by the complete denoising trajectory, avoiding approximation bias from intermediate-state evaluation.

## 4. SES: Spectral Evolution Search

To mitigate the curse of dimensionality in latent search, inference-time optimization should focus on a compact set of directions that exert strong control over the final generation. As shown in Figure 1, low-frequency perturbations significantly reshape the image structure, whereas high-frequency variations have negligible visual impact. Motivated by this observation, we explicitly constrain the search space to the low-frequency subspace.

In this section, we introduce *Spectral Evolution Search (SES)*, an inference-time scaling framework for initial noise optimization. As illustrated in Figure 2, SES consists of two key components: (1) wavelet-based spectral decoupling, which constructs a compact low-frequency search space; and (2) cross-entropy optimization within this subspace, which efficiently searches for high-reward regions under a limited inference budget.

### 4.1. Wavelet-based Spectral Decoupling

**Low-Frequency Search Space Construction.** Our goal is to construct a low-dimensional search space that retains the modes with the strongest control over the final generation. Unlike Fourier transforms, which provide global frequency decomposition but lack spatial locality, we employ the *Discrete Wavelet Transform (DWT)* to separate frequency bands while preserving coarse spatial structure.

Formally, consider an initial noise $\mathbf{x}_{\text{init}} \in \mathbb{R}^{C \times H \times W}$ sampled from $\mathcal{N}(\mathbf{0}, \mathbf{I})$. We apply a $J$-level orthogonal DWT $\mathcal{W}$ to decompose it into spectral components:

$$\mathbf{c} = \mathcal{W}(\mathbf{x}_{\text{init}}) = \left\{ \mathbf{c}_{LL}^{(J)}, \mathbf{c}_H^{\text{fixed}} \right\}. \tag{2}$$

Here, $\mathbf{c}_{LL}^{(J)}$ denotes the coarse low-frequency coefficients at level $J$, which encode global structure, while $\mathbf{c}_H^{\text{fixed}} = \{\mathcal{H}^{(1)}, \ldots, \mathcal{H}^{(J)}\}$ collects the high-frequency coefficients across all scales.

To exploit the stronger influence of low frequencies on the final generation, we keep the high-frequency component $\mathbf{c}_H^{\text{fixed}}$ fixed and optimize exclusively over the low-frequency vector $\mathbf{u} \triangleq \mathbf{c}_{LL}^{(J)} \in \mathbb{R}^{D'}$, where

$$D' = C \cdot (H/2^J) \cdot (W/2^J). \tag{3}$$

This reduces the search dimensionality by a factor of $4^J$, substantially alleviating the curse of dimensionality in noise-space optimization. Moreover, because the DWT is orthogonal, the low-frequency variable $\mathbf{u}$ follows a standard Gaussian marginal distribution. By fixing the high-frequency coefficients, the search remains within a valid affine subspace of the original prior, keeping the optimization compatible with the pre-trained distribution.

**Reconstruction to Noise Space.** During search, any candidate low-frequency vector $\mathbf{u}$ is recombined with the fixed $\mathbf{c}_H^{\text{fixed}}$ to form a valid initial noise map $\mathbf{x}_0(\mathbf{u})$. This recon-

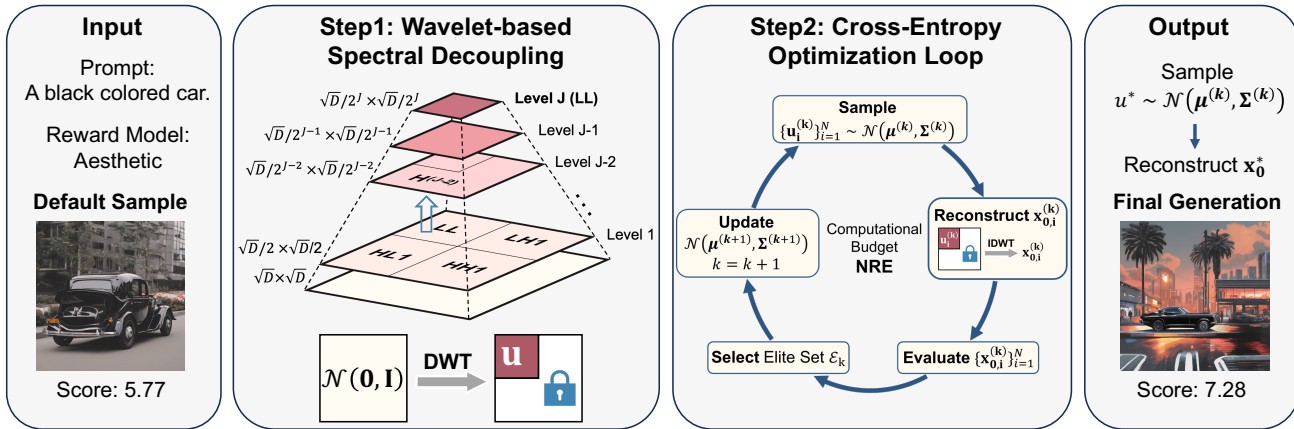

*Figure 2.* Overview of Spectral Evolution Search (SES). SES achieves inference-time scaling by optimizing the low-frequency components of the initial noise. First, SES performs wavelet-based spectral decoupling, freezing high-frequency components and constructing a low-frequency search space $\mathbf{u}$, thereby reducing the search dimension from $D$ to $D/4^J$. Subsequently, it executes a cross-entropy optimization loop, iteratively optimizing the distribution parameters $(\boldsymbol{\mu}, \boldsymbol{\Sigma})$ through a "Sample-Evaluate-Update" cycle within a limited budget (NRE). Finally, the optimal noise $\mathbf{x}^*$ is sampled and reconstructed to generate high-quality images with superior alignment.

struction is performed via the inverse DWT (IDWT) $\mathcal{W}^{-1}$:

$$\mathbf{x}_0(\mathbf{u}) = \mathcal{W}^{-1}\left(\mathbf{u} \oplus \mathbf{c}_H^{\text{fixed}}\right), \tag{4}$$

where $\oplus$ denotes the concatenation of wavelet coefficients. The reconstructed $\mathbf{x}_0(\mathbf{u})$ then serves as the initial noise for the sampler, enabling efficient reward evaluation of the candidate $\mathbf{u}$ during search.

### 4.2. Cross-Entropy Optimization on Low Frequencies

**Objective Formulation.** Having constructed the low-frequency search space, we formulate inference-time alignment as a black-box optimization problem. Our goal is to find the optimal low-frequency vector $\mathbf{u}^*$ that maximizes the expected reward $\mathcal{R}$:

$$\mathbf{u}^* = \underset{\mathbf{u} \in \mathbb{R}^{D'}}{\arg\max} \, \mathcal{R}\left(\Psi_\theta\left(\mathbf{x}_0(\mathbf{u}), c\right)\right). \tag{5}$$

Direct optimization of this objective presents two challenges: (1) the reward function $\mathcal{R}$ is typically non-differentiable; and (2) even with differentiable rewards, backpropagation through long-horizon denoising trajectories incurs substantial memory cost and gradient instability. We therefore employ the *Cross-Entropy Method (CEM)* as a gradient-free evolutionary strategy during inference.

CEM maintains a Gaussian search distribution $p(\mathbf{u}; \boldsymbol{\mu}, \boldsymbol{\Sigma})$ with diagonal covariance over the low-frequency variable $\mathbf{u}$. It iteratively steers the distribution towards high-reward regions through a "Sample-Evaluate-Update" loop. The detailed procedure is as follows:

**(1) Prior-Matched Initialization.** We initialize the distribution parameters as $\boldsymbol{\mu}^{(0)} = \mathbf{0}$ and $\boldsymbol{\Sigma}^{(0)} = \mathbf{I}$, parameterizing $\boldsymbol{\Sigma}$ as a diagonal matrix $\text{diag}(\boldsymbol{\sigma}^2)$. This aligns the initial search distribution with the pre-trained diffusion prior, reducing the risk of out-of-distribution initialization. We also initialize the candidate pool as $\mathcal{P} = \emptyset$.

**(2) Monte Carlo Sampling & Evaluation.** In the $k$-th iteration, we draw $N$ candidates $\{\mathbf{u}_i^{(k)}\}_{i=1}^N$ from the current distribution $\mathcal{N}(\boldsymbol{\mu}^{(k)}, \boldsymbol{\Sigma}^{(k)})$. Each candidate is combined with the fixed high-frequency component $\mathbf{c}_H^{\text{fixed}}$ to reconstruct the full noise $\mathbf{x}_{0,i}^{(k)}$ according to Eq. 4, which is then denoised to obtain its reward score $S_i^{(k)}$. All pairs $(\mathbf{u}_i^{(k)}, S_i^{(k)})$ are added to $\mathcal{P}$.

**(3) Elite-Driven Distribution Shaping.** After the $k$-th sampling round, we sort the candidate pool $\mathcal{P}$ by reward score, retain the top-$K$ candidates as the elite set $\mathcal{E}_k$, and discard the rest. Subsequently, we update the search distribution parameters using the statistics of these elites to guide the distribution towards high-reward regions. To reduce estimation variance under a small sample size and prevent premature convergence, we employ a smoothing factor $\gamma$ for momentum updates:

$$\begin{aligned}
\boldsymbol{\mu}^{(k+1)} &= (1-\gamma)\hat{\boldsymbol{\mu}}_{\mathcal{E}_k} + \gamma\boldsymbol{\mu}^{(k)}, \\
\boldsymbol{\sigma}^{2(k+1)} &= (1-\gamma)\hat{\boldsymbol{\sigma}}_{\mathcal{E}_k}^2 + \gamma\boldsymbol{\sigma}^{2(k)}.
\end{aligned} \tag{6}$$

**(4) Iterative Optimization and Final Generation.** Steps (2) and (3) are repeated until $k \times N$ reaches the preset computational budget. A final low-frequency variable is then sampled from the optimized distribution $\mathcal{N}(\boldsymbol{\mu}^{(k)}, \boldsymbol{\Sigma}^{(k)})$ to generate the final image.

**Discussion: Implicit Regularization vs. Reward Hacking.** By strictly confining optimization to the low-frequency subspace, SES imposes an implicit regularization that substantially limits access to the high-frequency perturbations of-

ten associated with *out-of-distribution (OOD) reward hacking*. In contrast, gradient-based guidance methods (e.g., DNO (Tang et al., 2025)) may exploit imperceptible high-frequency artifacts to maximize reward, leading to visual artifacts and texture collapse. By limiting access to high-frequency adversarial directions, our geometric constraint helps SES achieve high alignment scores while preserving the semantic integrity and naturalness of the generated images (detailed analysis in Appendix E).

# 5. Theoretical Analysis: Why Do Low Frequencies Dominate?

Why is the generation process significantly more sensitive to low-frequency perturbations? In this section, we address this question by analyzing the perturbation propagation dynamics within continuous-time diffusion models. Detailed mathematical derivations are provided in Appendix A.

Our analysis reveals a fundamental asymmetry: while the initial latent space is isotropic, the generative flow manifests pronounced anisotropy in the frequency domain. Specifically, we derive that the sensitivity of the generation result to initial perturbations follows a *power-law decay* with respect to spatial frequency. This theoretical finding corroborates the empirical evidence in Figure 1, confirming that low-frequency components of the initial noise are the primary drivers of the generative outcome.

## 5.1. Perturbation Dynamics of Generative Flows

We model the generation process as a deterministic ODE over the time interval $t \in [0, 1]$:

$$\frac{d\mathbf{x}_t}{dt} = v_\theta(\mathbf{x}_t, t). \tag{7}$$

To quantify how an infinitesimal perturbation $\boldsymbol{\xi}_0$ of the initial noise $\mathbf{x}_0$ propagates to the final sample $\mathbf{x}_1$, we analyze the first-order variational dynamics:

$$\frac{d\boldsymbol{\xi}_t}{dt} = \mathbf{J}_v(\mathbf{x}_t, t)\boldsymbol{\xi}_t, \tag{8}$$

where $\mathbf{J}_v$ denotes the Jacobian of the velocity field with respect to the state, and thus fully determines the local evolution of perturbations.

For continuous-time generative models based on the interpolation path $\mathbf{x}_t = \alpha(t)\mathbf{x}_1 + \sigma(t)\mathbf{x}_0$, the Jacobian $\mathbf{J}_v$ decomposes into two competing components (derivation in Appendix A.2):

$$\mathbf{J}_v(\mathbf{x}_t, t) = \underbrace{\mu(t)\mathbf{J}_{\hat{x}}}_{\text{Signal Amplification}} + \underbrace{\nu(t)\mathbf{I}}_{\text{Noise Contraction}}, \tag{9}$$

where $\mathbf{J}_{\hat{x}} = \nabla_\mathbf{x}\hat{\mathbf{x}}_\theta(\mathbf{x}_t, t)$ is the Jacobian of the denoiser. The scalar coefficients are given by $\nu(t) = \frac{\dot{\sigma}}{\sigma} < 0$ and $\mu(t) = \dot{\alpha} - \frac{\dot{\sigma}\alpha}{\sigma} > 0$.

This decomposition reveals a competition between two fundamental forces in the generative flow. The term $\nu(t)\mathbf{I}$ induces *isotropic contraction*, uniformly shrinking perturbations in all directions to suppress noise. In contrast, $\mu(t)\mathbf{J}_{\hat{x}}$ drives *anisotropic amplification*, selectively strengthening perturbations aligned with data-relevant directions captured by the denoiser. Their dynamic interplay ultimately governs the frequency-dependent sensitivity of the generative flow to initial perturbations.

## 5.2. Spectral Scaling Prediction in Frequency Domain

Directly analyzing this anisotropy in the original high-dimensional state space is intractable. To make the perturbation analysis analytically tractable, we move to the frequency domain and adopt a theoretical model based on local linearization and approximate spectral decoupling.

**Decoupled Spectral Dynamics.** Let $\tilde{\xi}_t(\omega) \triangleq \mathcal{F}\{\boldsymbol{\xi}_t\}(\omega)$ denote the scalar Fourier component of the perturbation at frequency $\omega$. By substituting Eq. 9 into Eq. 8 and approximating the denoiser's Jacobian as diagonally dominant, we decouple the high-dimensional perturbation dynamics into a set of independent scalar equations:

$$\frac{d\tilde{\xi}_t(\omega)}{dt} = [\mu(t)h(\omega, t) + \nu(t)]\tilde{\xi}_t(\omega), \tag{10}$$

where $h(\omega, t)$ denotes the effective spectral response of the denoising network. Integrating this equation over the full generation process gives the cumulative gain $G(\omega)$, which quantifies the total amplification of an initial perturbation at frequency $\omega$:

$$G(\omega) = \exp\left(\int_0^1 [\mu(\tau)h(\omega, \tau) + \nu(\tau)]\,d\tau\right). \tag{11}$$

Since the geometric contraction term $\nu(\tau)$ acts uniformly across the spectrum, the frequency-dependent variation of the cumulative gain is governed exclusively by the spectral response function $h(\omega, t)$.

**The Denoiser as a Wiener Filter.** To characterize $h(\omega, t)$, we approximate the trained network locally as an ideal Minimum Mean Square Error (MMSE) estimator. Natural images are well known to exhibit a power-law spectral decay $\|\omega\|^{-\gamma}$ (typically $\gamma \approx 2$) (Field, 1987). We therefore model the spectrum of the clean endpoint $\mathbf{x}_1$ as inheriting this heavy-tailed structure after encoding, namely $P_{\mathbf{x}_1}(\omega) \propto \|\omega\|^{-\beta}$, in contrast to the flat spectrum of Gaussian noise. This, in turn, induces a frequency-dependent Signal-to-Noise Ratio (SNR):

$$\text{SNR}(\omega, t) \propto \frac{\alpha^2(t)}{\sigma^2(t)} \cdot \|\omega\|^{-\beta}. \tag{12}$$

Under the MMSE objective, the optimal denoiser behaves as a frequency-domain Wiener filter (derivation in Appendix

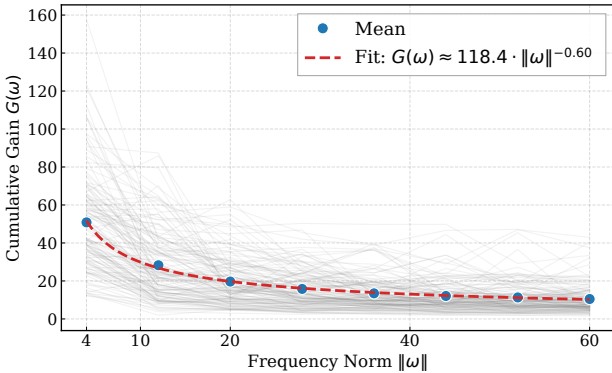

*Figure 3.* Validating the Spectral Scaling Prediction. We partition the frequency domain into 8 radial sub-bands and measure the cumulative gain of unit-norm perturbations ($\|\boldsymbol{\xi}\|_2 = 1$) injected into the SDXL initial noise. The results (blue dots, averaged over 100 prompts) reveal a monotonic decay in sensitivity as frequency increases. The strong alignment with the predicted power-law fit (red dashed line) validates the Spectral Scaling Prediction.

A.4), preserving high-SNR (low-frequency) components while suppressing low-SNR (high-frequency) noise:

$$h(\omega, t) = \frac{1}{\alpha(t)} \cdot \frac{\text{SNR}(\omega, t)}{\text{SNR}(\omega, t) + 1}. \tag{13}$$

Substituting this spectral response into the expression for cumulative gain yields our main theoretical prediction.

**Proposition 5.1** (Spectral Scaling Prediction; proof in Appendix A.5). *Under the assumptions of approximate spectral decoupling and local MMSE optimality, the cumulative gain $G(\omega)$ of an initial perturbation follows an inverse power-law scaling with respect to the frequency norm $\|\omega\|$:*

$$G(\omega) \propto \|\omega\|^{-\beta/2}, \tag{14}$$

*where $\beta$ denotes the spectral decay exponent of the data distribution in latent space (typically $0 < \beta < 2$).*

Proposition 5.1 reveals a central spectral bias in initial noise optimization: the effective degrees of freedom are intrinsically sparse and concentrated in the low-frequency band. As empirically corroborated in Figure 3, high-frequency perturbations exhibit negligible cumulative gain compared with low-frequency modes, rendering them ineffective directions for optimization. This theoretical result directly motivates the design of SES (Section 4) and justifies concentrating limited inference budget on low-frequency modes to maximize search efficiency.

## 6. Experiments

We provide a comprehensive evaluation of SES across multiple generative architectures and alignment objectives. Our experiments are designed to answer three questions: (1) how

effective SES is under fixed inference budgets; (2) which mechanisms are responsible for its performance gains; and (3) whether SES generalizes robustly across training-time aligned models and black-box reward functions. Detailed experimental settings and additional analyses are provided in Appendices C and D, respectively.

**Tasks and Rewards.** To evaluate the universality of SES, we consider four representative text-to-image models spanning two major generative paradigms: the latent-diffusion models Stable Diffusion (SD) v1.5 and SDXL (Podell et al., 2024), and the flow-matching models FLUX.1-dev (Labs, 2024) and Qwen-Image (Wu et al., 2025). For evaluation prompts, we use DrawBench (Saharia et al., 2022) for the SD series, and randomly sample 200 prompts from Pick-a-Pic (Kirstain et al., 2023) for the flow-matching models. We assess alignment under diverse reward objectives, including semantic consistency (CLIP Score (Hessel et al., 2021)), human preference (PickScore (Kirstain et al., 2023), HPSv2 (Wu et al., 2023), and ImageReward (Xu et al., 2023)), and aesthetic quality (Aesthetic Score).

**Baselines.** We compare SES against representative inference-time scaling strategies, including Best-of-N (BoN), Zero-Order Search (ZO-N), and Search over Paths (SoP) (Ma et al., 2025). We also include baselines based on particle filtering and resampling, such as SMC (Kim et al., 2025), SVDD (Li et al., 2025b), and Demon (Yeh et al., 2025). However, these methods rely on intermediate stochastic injections characteristic of SDE solvers. Since FLUX.1-dev and Qwen-Image use deterministic ODE solvers without stochastic injections during sampling, these baselines are incompatible and thus excluded from the flow-matching comparisons.

For fair comparison between methods that optimize initial noise and those that optimize intermediate latents, we adopt full denoising evaluation: every candidate is decoded to the final image space $\mathbf{x}_1$ before scoring. This avoids approximation errors caused by intermediate reward estimation and standardizes reward evaluation across methods. Unless otherwise specified, experiments use NRE = 200 and $T_{\text{total}} = 50$.

### 6.1. Inference-Time Scaling Performance

**Baseline Comparison.** We first compare SES against representative inference-time search strategies under a fixed computational budget (NRE = 200). As detailed in Table 1, SES consistently achieves the best reward scores across all settings, significantly outperforming BoN and trajectory-based approaches. Qualitative comparisons in Figure 4 further show that SES can generate high-fidelity samples even under highly non-convex human-preference objectives. Notably, SES maintains a clear advantage across both latent-diffusion and flow-matching architectures, underscoring its

*Table 1.* Quantitative comparison of inference-time scaling strategies on SDXL and FLUX.1-dev. We report the final reward scores achieved under a fixed budget of NRE = 200. Results are averaged over 5 random seeds and reported as mean$_{\pm\text{std}}$. "Baseline" denotes standard inference without search. Each column corresponds to a distinct experiment where the indicated metric serves as the sole optimization objective. The best results are highlighted in **bold**.

| Method | SDXL | | | | | FLUX.1-dev | | | | |
|---|---|---|---|---|---|---|---|---|---|---|
| | CLIP↑ | Pick↑ | HPSv2↑ | ImgRew.↑ | Aes.↑ | CLIP↑ | Pick↑ | HPSv2↑ | ImgRew.↑ | Aes.↑ |
| Baseline | $32.07_{\pm1.75}$ | $21.16_{\pm0.51}$ | $27.34_{\pm0.51}$ | $-0.14_{\pm0.26}$ | $5.22_{\pm0.16}$ | $33.26_{\pm0.86}$ | $22.00_{\pm0.41}$ | $28.08_{\pm0.26}$ | $1.02_{\pm0.13}$ | $6.19_{\pm0.13}$ |
| BoN | $43.30_{\pm0.39}$ | $23.77_{\pm0.06}$ | $30.86_{\pm0.06}$ | $1.54_{\pm0.03}$ | $6.35_{\pm0.06}$ | $39.50_{\pm0.12}$ | $23.17_{\pm0.09}$ | $30.10_{\pm0.11}$ | $1.67_{\pm0.03}$ | $6.80_{\pm0.07}$ |
| ZO-N | $41.30_{\pm0.27}$ | $22.98_{\pm0.21}$ | $30.55_{\pm0.50}$ | $1.31_{\pm0.09}$ | $6.21_{\pm0.05}$ | $38.10_{\pm0.18}$ | $22.95_{\pm0.21}$ | $29.62_{\pm0.25}$ | $1.54_{\pm0.05}$ | $6.57_{\pm0.11}$ |
| SoP | $40.74_{\pm0.18}$ | $22.95_{\pm0.11}$ | $29.85_{\pm0.21}$ | $1.20_{\pm0.09}$ | $5.92_{\pm0.10}$ | $37.27_{\pm0.29}$ | $22.63_{\pm0.24}$ | $29.52_{\pm0.19}$ | $1.35_{\pm0.02}$ | $6.45_{\pm0.14}$ |
| SMC | $42.53_{\pm0.20}$ | $23.59_{\pm0.18}$ | $30.71_{\pm0.20}$ | $1.40_{\pm0.07}$ | $6.36_{\pm0.17}$ | - | - | - | - | - |
| SVDD | $41.50_{\pm0.37}$ | $23.30_{\pm0.29}$ | $30.35_{\pm0.37}$ | $1.38_{\pm0.13}$ | $6.23_{\pm0.19}$ | - | - | - | - | - |
| Demon | $41.81_{\pm0.52}$ | $23.82_{\pm0.14}$ | $31.03_{\pm0.19}$ | $1.48_{\pm0.07}$ | $6.34_{\pm0.08}$ | - | - | - | - | - |
| **SES** | $\mathbf{43.53}_{\pm0.54}$ | $\mathbf{23.97}_{\pm0.06}$ | $\mathbf{31.45}_{\pm0.26}$ | $\mathbf{1.62}_{\pm0.02}$ | $\mathbf{6.55}_{\pm0.06}$ | $\mathbf{40.18}_{\pm0.19}$ | $\mathbf{23.35}_{\pm0.15}$ | $\mathbf{30.68}_{\pm0.17}$ | $\mathbf{1.79}_{\pm0.01}$ | $\mathbf{7.03}_{\pm0.08}$ |

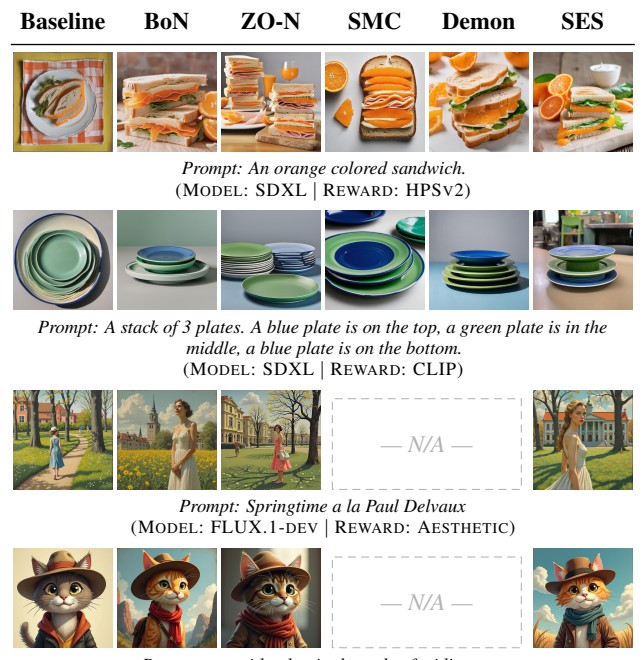

| Baseline | BoN | ZO-N | SMC | Demon | SES |

*Prompt: An orange colored sandwich.*
(MODEL: SDXL | REWARD: HPSv2)

*Prompt: A stack of 3 plates. A blue plate is on the top, a green plate is in the middle, a blue plate is on the bottom.*
(MODEL: SDXL | REWARD: CLIP)

— N/A —

*Prompt: Springtime a la Paul Delvaux*
(MODEL: FLUX.1-DEV | REWARD: AESTHETIC)

— N/A —

*Prompt: cat with a hat in the style of midjourney*
(MODEL: FLUX.1-DEV | REWARD: PICKSCORE)

*Figure 4.* Qualitative comparison under a fixed budget (NRE = 200). Visual samples generated by SES and competing baselines across different base models (SDXL, FLUX.1-dev) and reward objectives, demonstrating SES's superior reward alignment.

strong cross-architecture generalization. Results on SD v1.5 and Qwen-Image are provided in Appendix D.1.

**Scaling Behavior.** We further investigate the scaling properties of SES by extending the computational budget to NRE = 1000. As shown in Figure 5, Best-of-N exhibits roughly logarithmic scaling and quickly plateaus once NRE > 200. In contrast, SES maintains strong performance growth throughout the tested range, with no clear sign of saturation. Furthermore, SES consistently outperforms trajectory-based methods across the entire budget

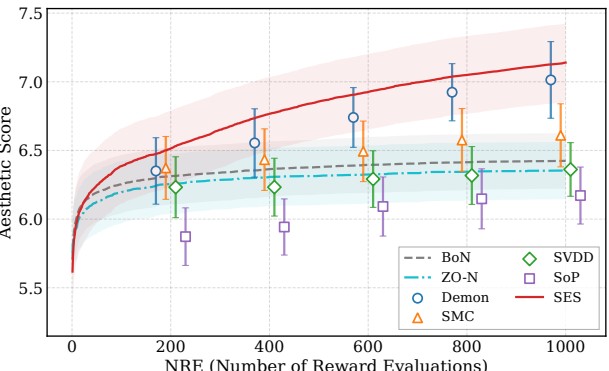

*Figure 5.* Scaling behavior analysis. We investigate scaling behaviors by extending the computational budget to NRE = 1000, using SDXL with Aesthetic Score. Curves depict the continuous scaling trajectory, while markers indicate performance measured at discrete checkpoints (NRE ∈ {200, 400, 600, 800, 1000}). SES exhibits continuous performance gains.

spectrum. This confirms that restricting the search to the low-frequency subspace concentrates the computational budget on the most influential degrees of freedom, enabling more effective exploration compared to indiscriminate full-space sampling.

### 6.2. Mechanism Analysis and Efficiency

**Ablation Studies.** We conduct an ablation study on SDXL (Table 2) to identify the main factors behind SES's performance gains. *Subspace Selection.* Restricting search to the low-frequency subspace yields consistently better results than searching over the full-frequency space or only the high-frequency subspace. Notably, it also substantially outperforms a random subspace of the same dimensionality. This indicates that SES benefits not merely from dimensionality reduction, but from choosing a subspace whose directions exert stronger influence on the final generation. *Decomposition Granularity.* Performance is strongest at intermediate wavelet decomposition levels (e.g., $J = 3, 4, 5$),

*Table 2.* Ablation analysis on SDXL targeting Aesthetic Score (NRE = 200). We systematically evaluate three key components against the **Default** configuration (LL Subband, Level $J$=4, CEM): (1) spectral subspace selection (Top), (2) wavelet decomposition levels $J$ (Middle), and (3) search strategies (Bottom).

| Ablation Setting | Aesthetic ↑ | Δ |
|---|---|---|
| **Default** | **6.49** | – |
| *1. Subspace Selection* | | |
| High Freq. (LH+HL+HH) | 5.84 | -0.65 |
| Full Frequency Space | 5.96 | -0.53 |
| Random Subspace | 5.66 | -0.83 |
| *2. Decomposition Level* | | |
| Level 2 | 6.38 | -0.11 |
| Level 3 | 6.48 | -0.01 |
| Level 5 | 6.47 | -0.02 |
| Level 6 | 6.29 | -0.20 |
| *3. Optimization Strategy* | | |
| Random Search | 6.28 | -0.21 |

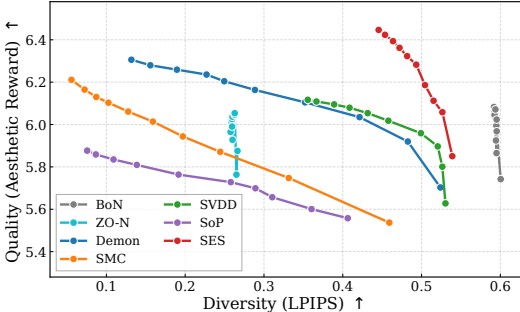

*Figure 6.* Quality-diversity trade-off analysis on SDXL. We visualize the optimization trajectory using Average Aesthetic Score (Quality) versus Pairwise LPIPS (Diversity). Markers along each curve represent snapshots at intervals of 20 NRE, ranging from NRE = 20 to 200. This demonstrates that SES effectively balances generation quality and diversity.

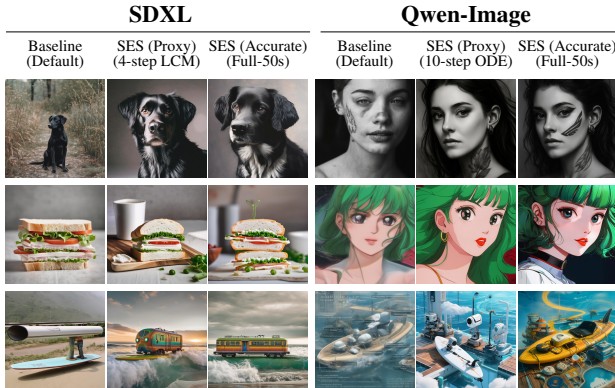

*Figure 7.* Qualitative comparison: Proxy vs. Accurate reward guidance targeting Aesthetic Score. Left (SDXL): Proxy via 4-step LCM. Right (Qwen-Image): Proxy via 10-step ODE, both targeting the Aesthetic Score. The high structural consistency between the two settings confirms that SES effectively leverages low-cost proxies to locate the optimal generation pattern.

while both overly shallow and overly aggressive decompositions are less effective. This reflects a trade-off: when $J$ is too small, the retained subspace remains relatively high-dimensional and still includes weakly influential directions; when $J$ is too large, the search space becomes overly compressed and loses important structural degrees of freedom. *Optimization Strategy.* With the search space fixed, CEM provides a clear improvement over random search. This shows that selecting the right subspace is necessary but not sufficient: effective optimization within that subspace is also important for realizing the full benefit of SES. Overall, the results suggest that the low-frequency subspace provides a particularly effective search space, concentrating the most sensitive directions while remaining amenable to efficient evolutionary optimization.

**Quality-Diversity Trade-off.** We analyze the quality-diversity trade-off through LPIPS-reward trajectories (Figure 6). Because particle filtering methods rely on hard resampling, they are prone to sample impoverishment, which leads to premature convergence and mode collapse. In contrast, SES performs smooth distributional updates within the low-frequency subspace, allowing the population to move gradually toward high-reward regions while preserving diversity. As a result, SES achieves a more favorable quality-diversity trade-off than competing methods.

**Accelerated Search via Proxy Guidance.** To evaluate rewards, latent candidates must be decoded into image space. To reduce the latency of standard iterative decoding, we introduce a proxy evaluation mechanism: 4-step Latent Consistency Model (LCM) decoding (Luo et al., 2023) for latent-diffusion models, and 10-step ODE integration for flow-matching models. The effectiveness of this strategy

relies on ranking consistency. Although proxy samples are generated by simplified decoding paths, they preserve sufficient semantic fidelity to maintain the relative ranking of candidates, which is sufficient for rank-based evolutionary search. As shown in Figure 8, this strategy reduces evaluation cost by approximately 85%, while SES continues to outperform baselines on SDXL under the same proxy-guidance setting. Qualitative results in Figure 7 further show that proxy-guided SES produces samples that outperform baseline methods, while remaining structurally consistent with results obtained under full reward evaluation. Results on FLUX.1-dev are provided in Appendix D.2.

### 6.3. Generalization and Robustness

**Complementarity with Training-Time Alignment.** An important question is whether inference-time scaling only compensates for deficiencies in unaligned models, or whether it can further improve already aligned ones. To examine this,

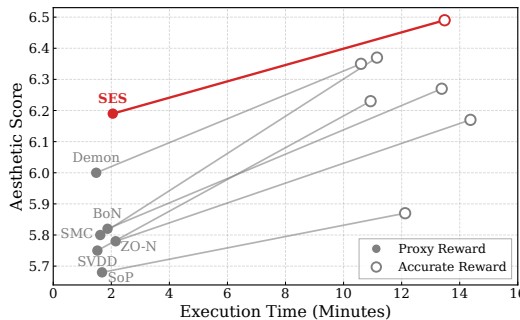

*Figure 8.* Efficiency analysis on SDXL comparing Accurate vs. Proxy guidance. The adoption of Proxy evaluation reduces execution time by ∼85%, and SES consistently outperforms all baselines under identical evaluation settings.

*Table 3.* Orthogonality to training-time alignment. Even with strong initial alignment from DPO and SPO, SES achieves consistent gains across all metrics. Best results are highlighted in **bold**.

| Method | CLIP | PickScore | HPSv2 | Aes. | ImgRew. |
|---|---|---|---|---|---|
| SDXL | 33.05 | 21.76 | 27.64 | 5.48 | 0.33 |
| + DPO | 36.77 | 22.48 | 29.02 | 5.59 | 0.77 |
| + DPO & SES | **43.67** | **23.94** | **31.63** | **6.40** | **1.61** |
| + SPO | 33.12 | 22.59 | 28.69 | 5.90 | 0.35 |
| + SPO & SES | **41.18** | **24.35** | **31.35** | **6.78** | **1.65** |

we apply SES to models fine-tuned with Direct Preference Optimization (DPO) (Wallace et al., 2024) and Step-by-Step Preference Optimization (SPO) (Liang et al., 2025). As shown in Table 3, SES delivers consistent gains even on models that have already undergone alignment tuning. This suggests that the low-frequency search space remains effective after training-time alignment, and that SES complements training-time alignment methods rather than merely compensating for missing alignment.

**Generalization across Rewards and Samplers.** Because SES is gradient-free, it is broadly compatible with black-box reward functions, including non-differentiable Vision-Language Models (VLMs) (Appendix D.4). Moreover, since SES optimizes the initial noise rather than intervening in the denoising trajectory, it remains compatible with a wide range of samplers. Results in Appendix D.3 show robust performance across both deterministic ODE solvers and stochastic SDE samplers.

## 7. Conclusion

We presented Spectral Evolution Search (SES), a model-agnostic inference-time scaling framework for initial noise optimization. Motivated by our finding that final generations are substantially more sensitive to low-frequency perturbations in the initial noise, SES restricts search to a compact low-frequency subspace via wavelet-based spectral decoupling, and then applies the Cross-Entropy Method (CEM)

to efficiently explore this space under limited inference budgets. This design mitigates the curse of dimensionality that hampers isotropic search in high-dimensional noise spaces.

Our theoretical analysis explains this design through perturbation propagation dynamics, yielding the Spectral Scaling Prediction, which states that low-frequency perturbations enjoy substantially larger cumulative gain than high-frequency ones. Extensive experiments across diffusion and flow-matching architectures, diverse reward objectives, and multiple sampling settings demonstrate that SES delivers strong and robust inference-time scaling performance.

**Limitations.** Like other inference-time scaling strategies, SES depends on the quality and cost of the reward model. Theoretically, our analysis relies on local linearization and approximate spectral decoupling, thus not capturing higher-order interactions. Practically, although NRE provides a hardware-agnostic measure of inference budget, it does not necessarily translate linearly to wall-clock latency.

## Acknowledgements

This work was supported by the Guizhou Provincial Program on Commercialization of Scientific and Technological Achievements (Qiankehezhongyindi [2025] No. 006) and Alibaba Group through the Alibaba Innovation Research Program.

## Impact Statement

This paper presents work whose goal is to advance the field of Machine Learning. There are many potential societal consequences of our work, none of which we feel must be specifically highlighted here.

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

## Appendix Contents

## A. Proofs for Theoretical Analysis

### A.1. Defining the Difference Calculation Objective

The baseline point is the initial noise $\mathbf{x}_0$. We add a perturbation $\boldsymbol{\xi}_0$ to obtain the perturbed initial noise $\tilde{\mathbf{x}}_0 = \mathbf{x}_0 + \boldsymbol{\xi}_0$. We aim to determine how this perturbation affects the final generated image, which entails calculating the difference between the generated data corresponding to the two initial noise inputs: $\boldsymbol{\xi}_1 = \tilde{\mathbf{x}}_1 - \mathbf{x}_1$.

Both $\mathbf{x}_t$ and $\tilde{\mathbf{x}}_t$ are solutions to $\mathrm{d}\mathbf{x}_t = v_\theta(\mathbf{x}_t, t)\mathrm{d}t$, satisfying the integral equations:

$$\mathbf{x}_t = \mathbf{x}_0 + \int_0^t v_\theta(\mathbf{x}_\tau, \tau)\,\mathrm{d}\tau \tag{15}$$

$$\tilde{\mathbf{x}}_t = (\mathbf{x}_0 + \boldsymbol{\xi}_0) + \int_0^t v_\theta(\tilde{\mathbf{x}}_\tau, \tau)\,\mathrm{d}\tau. \tag{16}$$

Directly calculating $\boldsymbol{\xi}_1$ requires evaluating:

$$\boldsymbol{\xi}_1 = \tilde{\mathbf{x}}_1 - \mathbf{x}_1 = \boldsymbol{\xi}_0 + \int_0^1 \left(v_\theta(\tilde{\mathbf{x}}_\tau, \tau) - v_\theta(\mathbf{x}_\tau, \tau)\right)\,\mathrm{d}\tau. \tag{17}$$

Here, the term $v_\theta(\tilde{\mathbf{x}}_\tau, \tau) - v_\theta(\mathbf{x}_\tau, \tau)$ depends on the unknown trajectories $\tilde{\mathbf{x}}_\tau$ and $\mathbf{x}_\tau$. Since $v_\theta$ is a highly non-linear function, it is infeasible to directly derive an explicit expression for $\boldsymbol{\xi}_t$.

To address this, we analyze the rate of change of the difference, $\frac{\mathrm{d}\boldsymbol{\xi}_t}{\mathrm{d}t}$. If this value is positive, the difference diverges at time $t$; otherwise, it converges.

$$\frac{\mathrm{d}\boldsymbol{\xi}_t}{\mathrm{d}t} = \frac{\mathrm{d}(\mathbf{x}_t + \boldsymbol{\xi}_t)}{\mathrm{d}t} - \frac{\mathrm{d}\mathbf{x}_t}{\mathrm{d}t} = v_\theta(\mathbf{x}_t + \boldsymbol{\xi}_t, t) - v_\theta(\mathbf{x}_t, t). \tag{18}$$

Performing a first-order Taylor expansion on $v_\theta(\mathbf{x}_t + \boldsymbol{\xi}_t, t)$, we obtain:

$$v_\theta(\mathbf{x}_t + \boldsymbol{\xi}_t, t) \approx v_\theta(\mathbf{x}_t, t) + \nabla_\mathbf{x} v_\theta(\mathbf{x}_t, t) \cdot \boldsymbol{\xi}_t + O(\|\boldsymbol{\xi}_t\|^2). \tag{19}$$

Neglecting higher-order terms, the evolution is governed by:

$$\frac{\mathrm{d}\boldsymbol{\xi}_t}{\mathrm{d}t} \approx \nabla_\mathbf{x} v_\theta(\mathbf{x}_t, t) \cdot \boldsymbol{\xi}_t = \mathbf{J}_v(\mathbf{x}_t, t)\boldsymbol{\xi}_t, \tag{20}$$

where $\mathbf{J}_v(\mathbf{x}_t, t)$ is the instantaneous Jacobian matrix of the velocity field.

### A.2. Derivation of the Jacobian of the Velocity Field

In this section, we derive the specific form of $\mathbf{J}_v$. In mainstream diffusion and flow-matching models, the forward process satisfies the following trajectory interpolation formula:

$$\mathbf{x}_t = \alpha(t)\mathbf{x}_1 + \sigma(t)\mathbf{x}_0, \quad t \in [0, 1], \tag{21}$$

where $\alpha(t)$ and $\sigma(t)$ are time-dependent scheduling functions. Differentiating both sides with respect to time $t$ yields the ideal flow field:

$$\frac{\mathrm{d}\mathbf{x}_t}{\mathrm{d}t} = \dot{\alpha}(t)\mathbf{x}_1 + \dot{\sigma}(t)\mathbf{x}_0. \tag{22}$$

During inference, the model employs a neural network $\hat{\mathbf{x}}_\theta(\mathbf{x}_t, t)$ to estimate $\mathbf{x}_1$, allowing us to solve for the estimate of $\mathbf{x}_0$:

$$\mathbf{x}_0 = \frac{\mathbf{x}_t - \alpha(t)\hat{\mathbf{x}}_\theta(\mathbf{x}_t, t)}{\sigma(t)}. \tag{23}$$

Substituting this estimate of $\mathbf{x}_0$ into the derivative formula and replacing $\mathbf{x}_1$ with $\hat{\mathbf{x}}_\theta$, we obtain the parameterized velocity field $v_\theta$:

$$\begin{aligned}
v_\theta(\mathbf{x}_t, t) &= \dot{\alpha}(t)\hat{\mathbf{x}}_\theta(\mathbf{x}_t, t) + \dot{\sigma}(t)\left(\frac{\mathbf{x}_t - \alpha(t)\hat{\mathbf{x}}_\theta(\mathbf{x}_t, t)}{\sigma(t)}\right) \\
&= \dot{\alpha}(t)\hat{\mathbf{x}}_\theta(\mathbf{x}_t, t) + \frac{\dot{\sigma}(t)}{\sigma(t)}\mathbf{x}_t - \frac{\dot{\sigma}(t)\alpha(t)}{\sigma(t)}\hat{\mathbf{x}}_\theta(\mathbf{x}_t, t) \\
&= \left(\dot{\alpha}(t) - \frac{\dot{\sigma}(t)\alpha(t)}{\sigma(t)}\right)\hat{\mathbf{x}}_\theta(\mathbf{x}_t, t) + \frac{\dot{\sigma}(t)}{\sigma(t)}\mathbf{x}_t.
\end{aligned} \tag{24}$$

We define two time-varying coefficients $\mu(t)$ and $\nu(t)$:

$$\mu(t) \triangleq \dot{\alpha}(t) - \frac{\dot{\sigma}(t)\alpha(t)}{\sigma(t)}, \quad \nu(t) \triangleq \frac{\dot{\sigma}(t)}{\sigma(t)}. \tag{25}$$

Thus, the parameterized velocity field simplifies to:

$$v_\theta(\mathbf{x}_t, t) = \mu(t)\hat{\mathbf{x}}_\theta(\mathbf{x}_t, t) + \nu(t)\mathbf{x}_t. \tag{26}$$

Consequently, the specific form of the Jacobian matrix of the velocity field is:

$$\begin{aligned}
\mathbf{J}_v(\mathbf{x}_t, t) &= \nabla_\mathbf{x}\left(\mu(t)\hat{\mathbf{x}}_\theta(\mathbf{x}_t, t) + \nu(t)\mathbf{x}_t\right) \\
&= \mu(t)\nabla_\mathbf{x}\hat{\mathbf{x}}_\theta(\mathbf{x}_t, t) + \nu(t)\nabla_\mathbf{x}\mathbf{x}_t \\
&= \mu(t)\mathbf{J}_{\hat{x}} + \nu(t)\mathbf{I},
\end{aligned} \tag{27}$$

where $\mathbf{J}_{\hat{x}} \triangleq \nabla_\mathbf{x}\hat{\mathbf{x}}_\theta(\mathbf{x}_t, t)$ is the input-output Jacobian matrix of the neural network, and $\mathbf{I}$ is the identity matrix. The specific form governing the rate of error change is:

$$\frac{\mathrm{d}\boldsymbol{\xi}_t}{\mathrm{d}t} = \mathbf{J}_v(\mathbf{x}_t, t)\boldsymbol{\xi}_t = (\mu(t)\mathbf{J}_{\hat{x}} + \nu(t)\mathbf{I})\boldsymbol{\xi}_t. \tag{28}$$

**A.3. Solution of Spectral Evolution Dynamics**

In this section, we detail how to decouple the high-dimensional variational equation derived in Appendix A.2 using frequency domain analysis and solve for the cumulative gain $G(\omega)$ of the initial perturbation.

Recall the linearized variational equation:

$$\frac{d\boldsymbol{\xi}_t}{dt} = \mu(t)\mathbf{J}_{\hat{x}}\boldsymbol{\xi}_t + \nu(t)\mathbf{I}\boldsymbol{\xi}_t. \tag{29}$$

To decouple the dependencies across spatial dimensions, we apply the spatial Fourier transform operator $\mathcal{F}$ to both sides. Let $\tilde{\xi}_t(\omega) \triangleq \mathcal{F}\{\boldsymbol{\xi}_t\}(\omega)$ denote the spectral component of the error vector $\boldsymbol{\xi}_t$ at frequency $\omega$.

Using the linearity of the Fourier transform, we process each term:

Left-hand side: Since the Fourier transform acts as an integral over spatial coordinates and the time derivative acts on the time coordinate, under appropriate smoothness conditions, the order of operations can be swapped:

$$\mathcal{F}\left\{\frac{d\boldsymbol{\xi}_t}{dt}\right\} = \frac{d\tilde{\xi}_t(\omega)}{dt}. \tag{30}$$

Right-hand side, first term: Based on the frequency domain decoupling assumption, the denoising network Jacobian $\mathbf{J}_{\hat{x}}$ is approximated as a diagonal operator in the frequency domain. This implies its action in the frequency domain is equivalent to scalar multiplication. We define the effective spectral response $h(\omega, t)$:

$$\mathcal{F}\{\mathbf{J}_{\hat{x}}\boldsymbol{\xi}\}(\omega) \approx h(\omega, t) \cdot \mathcal{F}\{\boldsymbol{\xi}\}(\omega). \tag{31}$$

Right-hand side, second term: Since the identity matrix $\mathbf{I}$ remains the identity operator in the frequency domain, and the coefficient $\nu(t)$ depends only on time:

$$\mathcal{F}\{\nu(t)\mathbf{I}\boldsymbol{\xi}_t\}(\omega) = \nu(t) \cdot \tilde{\xi}_t(\omega). \tag{32}$$

Accordingly, the complex high-dimensional matrix differential equation decouples into independent scalar Ordinary Differential Equations (ODEs) for each frequency $\omega$:

$$\frac{d\tilde{\xi}_t(\omega)}{dt} = (\mu(t)h(\omega, t) + \nu(t))\, \tilde{\xi}_t(\omega). \tag{33}$$

To simplify notation, we define the instantaneous spectral eigenvalue $\lambda(\omega, t)$ as:

$$\lambda(\omega, t) \triangleq \mu(t)h(\omega, t) + \nu(t). \tag{34}$$

Next, we solve this ODE using the separation of variables:

$$\frac{d\tilde{\xi}_t}{\tilde{\xi}_t} = \lambda(\omega, t)dt. \tag{35}$$

Integrating both sides over the time interval $t \in [0, 1]$:

$$\int_{\tilde{\xi}_0}^{\tilde{\xi}_1} \frac{d\tilde{\xi}}{\tilde{\xi}} = \int_0^1 \lambda(\omega, \tau)d\tau \implies \ln\left(\frac{\tilde{\xi}_1(\omega)}{\tilde{\xi}_0(\omega)}\right) = \int_0^1 \lambda(\omega, \tau)d\tau. \tag{36}$$

Exponentiating yields the cumulative gain $G(\omega)$ at frequency $\omega$:

$$G(\omega) \triangleq \frac{\|\tilde{\xi}_1(\omega)\|}{\|\tilde{\xi}_0(\omega)\|} = \exp\left(\int_0^1 (\mu(\tau)h(\omega, \tau) + \nu(\tau))\, d\tau\right). \tag{37}$$

## A.4. Calculation of Effective Spectral Response

We first derive the Signal-to-Noise Ratio (SNR) of the latent variable $\mathbf{x}_t$ in the frequency domain, and then use it to obtain the effective spectral response $h(\omega, t)$.

**Power Spectrum Decomposition of Latent Variables.** In the latent generative models considered in this work, the generation process operates in a low-dimensional latent space. At any time $t \in [0, 1]$, the latent variable $\mathbf{x}_t$ is formed by the linear superposition of the clean data latent code $\mathbf{x}_1$ and Gaussian noise $\mathbf{x}_0$:

$$\mathbf{x}_t = \alpha(t)\mathbf{x}_1 + \sigma(t)\mathbf{x}_0. \tag{38}$$

Here, $\mathbf{x}_0 \sim \mathcal{N}(\mathbf{0}, \mathbf{I})$. Assuming statistical independence between the clean latent code and the initial noise, the Power Spectral Density (PSD) of the mixed signal is the weighted sum of the component power spectra:

$$P_{\mathbf{x}_t}(\omega) = \alpha^2(t)P_{\mathbf{x}_1}(\omega) + \sigma^2(t)P_{\mathbf{x}_0}(\omega). \tag{39}$$

**Spectral Statistical Properties of Signal and Noise.** *Clean latent code* $\mathbf{x}_1$. Raw natural images exhibit strong spatial redundancy and correlation, and their power spectra typically follow a power-law decay with exponent close to 2 (Field, 1987). The latent code $\mathbf{x}_1$ is obtained through an encoder $\mathcal{E}$, i.e.,

$$\mathbf{x}_1 = \mathcal{E}(\text{Image}). \tag{40}$$

Due to downsampling and KL regularization, the encoder performs a partial form of spectral whitening: high-frequency redundancy is compressed, while the latent distribution is regularized toward a standard normal prior. As a result, the latent spectrum is flatter than that of raw images, but still retains a heavy-tailed low-frequency bias. We therefore assume that the radial power spectrum of $\mathbf{x}_1$ follows

$$P_{\mathbf{x}_1}(\omega) = \mathbb{E}\big[|\tilde{\mathbf{x}}_1(\omega)|^2\big] \propto \|\omega\|^{-\beta}, \tag{41}$$

where $\|\omega\| = \sqrt{\omega_h^2 + \omega_w^2}$ denotes the radial spatial frequency. Owing to the decorrelation effect of the latent space, the decay exponent typically satisfies $0 < \beta < 2$.

*Gaussian noise* $\mathbf{x}_0$. Standard Gaussian noise $\mathbf{x}_0$ remains independent and identically distributed (i.i.d.) in the discrete Fourier basis. By Parseval's theorem, its power spectrum is flat across frequencies:

$$P_{\mathbf{x}_0}(\omega) = \mathbb{E}\big[|\tilde{\mathbf{x}}_0(\omega)|^2\big] = 1. \tag{42}$$

**Derivation of Local SNR.** Under the independence assumption, the local SNR at frequency $\omega$ is defined as the ratio of signal power to noise power:

$$\text{SNR}(\omega, t) \triangleq \frac{\alpha^2(t)P_{\mathbf{x}_1}(\omega)}{\sigma^2(t)P_{\mathbf{x}_0}(\omega)}. \tag{43}$$

Substituting $P_{\mathbf{x}_1}(\omega) \propto \|\omega\|^{-\beta}$ and $P_{\mathbf{x}_0}(\omega) = 1$, we obtain

$$\text{SNR}(\omega, t) \propto \frac{\alpha^2(t)}{\sigma^2(t)} \cdot \|\omega\|^{-\beta}. \tag{44}$$

We next derive the effective spectral response $h(\omega, t)$.

Let $\mathcal{F}$ denote the Fourier transform operator, and let $\tilde{\mathbf{x}}_t(\omega)$, $\tilde{\mathbf{x}}_1(\omega)$, and $\tilde{\mathbf{x}}_0(\omega)$ denote the Fourier coefficients of $\mathbf{x}_t$, $\mathbf{x}_1$, and $\mathbf{x}_0$ at frequency $\omega$, respectively. Under the approximate frequency-domain decoupling assumption, the local action of the denoiser $\hat{\mathbf{x}}_\theta$ at frequency $\omega$ can be modeled as a scalar gain:

$$\mathcal{F}\{\hat{\mathbf{x}}_\theta(\mathbf{x}_t)\}(\omega) \approx h(\omega, t)\,\tilde{\mathbf{x}}_t(\omega). \tag{45}$$

We determine the optimal gain $h(\omega, t)$ by minimizing the mean squared error between the denoiser output and the clean target at that frequency:

$$\min_h \; \mathcal{L}(h) = \mathbb{E}\Big[\|h\,\tilde{\mathbf{x}}_t(\omega) - \tilde{\mathbf{x}}_1(\omega)\|^2\Big]. \tag{46}$$

Since

$$\mathbf{x}_t = \alpha(t)\mathbf{x}_1 + \sigma(t)\mathbf{x}_0, \tag{47}$$

its Fourier representation satisfies

$$\tilde{\mathbf{x}}_t(\omega) = \alpha(t)\tilde{\mathbf{x}}_1(\omega) + \sigma(t)\tilde{\mathbf{x}}_0(\omega). \tag{48}$$

Substituting this into the loss and using the independence of $\mathbf{x}_1$ and $\mathbf{x}_0$ (so that the cross term vanishes in expectation), we obtain

$$
\begin{aligned}
\mathcal{L}(h) &= \mathbb{E}\Big[\big\|h\big(\alpha(t)\tilde{\mathbf{x}}_1(\omega) + \sigma(t)\tilde{\mathbf{x}}_0(\omega)\big) - \tilde{\mathbf{x}}_1(\omega)\big\|^2\Big] \\
&= \mathbb{E}\Big[\big\|(h\alpha(t) - 1)\tilde{\mathbf{x}}_1(\omega) + h\sigma(t)\tilde{\mathbf{x}}_0(\omega)\big\|^2\Big] \\
&= \big(h\alpha(t) - 1\big)^2 \underbrace{\mathbb{E}\big[\|\tilde{\mathbf{x}}_1(\omega)\|^2\big]}_{P_{\mathbf{x}_1}(\omega)} + h^2\sigma^2(t)\underbrace{\mathbb{E}\big[\|\tilde{\mathbf{x}}_0(\omega)\|^2\big]}_{P_{\mathbf{x}_0}(\omega)}.
\end{aligned}
\tag{49}
$$

Setting the derivative with respect to $h$ to zero gives

$$\frac{\mathrm{d}\mathcal{L}}{\mathrm{d}h} = 2\big(h\alpha(t) - 1\big)\alpha(t)P_{\mathbf{x}_1}(\omega) + 2h\sigma^2(t)P_{\mathbf{x}_0}(\omega) = 0. \tag{50}$$

Solving for $h$ yields

$$h(\omega, t) = \frac{\alpha(t)P_{\mathbf{x}_1}(\omega)}{\alpha^2(t)P_{\mathbf{x}_1}(\omega) + \sigma^2(t)P_{\mathbf{x}_0}(\omega)}. \tag{51}$$

Using the definition of $\mathrm{SNR}(\omega, t)$, we can rewrite this expression as

$$h(\omega, t) = \frac{1}{\alpha(t)} \cdot \frac{\mathrm{SNR}(\omega, t)}{\mathrm{SNR}(\omega, t) + 1}. \tag{52}$$

This is precisely the frequency-domain form of a Wiener filter: high-SNR components (typically low frequencies) are preserved, while low-SNR components (typically high frequencies) are attenuated.

### A.5. Derivation of the Spectral Scaling Prediction

In this section, we employ matched asymptotic analysis to determine the cumulative gain $G(\omega)$.

**Algebraic Rearrangement of the Instantaneous Growth Rate.** Recall the eigenvalue equation $\lambda(\omega, t) = \mu(t)h(\omega, t) + \nu(t)$, where $\mu = \dot{\alpha} - \frac{\dot{\sigma}\alpha}{\sigma}$ and $\nu = \frac{\dot{\sigma}}{\sigma}$. Substituting the optimal spectral response $h(\omega, t) = \frac{1}{\alpha(t)}\frac{\mathrm{SNR}(\omega,t)}{\mathrm{SNR}(\omega,t)+1}$ derived in Appendix A.4 into the equation yields:

$$
\begin{aligned}
\lambda(\omega, t) &= \left(\dot{\alpha} - \frac{\dot{\sigma}\alpha}{\sigma}\right)\frac{1}{\alpha}\rho(\omega, t) + \frac{\dot{\sigma}}{\sigma} \\
&= \frac{\dot{\alpha}}{\alpha}\rho(\omega, t) - \frac{\dot{\sigma}}{\sigma}\rho(\omega, t) + \frac{\dot{\sigma}}{\sigma} \\
&= \frac{\dot{\sigma}}{\sigma} + \left(\frac{\dot{\alpha}}{\alpha} - \frac{\dot{\sigma}}{\sigma}\right)\rho(\omega, t),
\end{aligned}
\tag{53}
$$

where the spectral weight function is defined as $\rho(\omega, t) = \frac{\mathrm{SNR}(\omega,t)}{\mathrm{SNR}(\omega,t)+1}$. Given that the generation process typically satisfies $\dot{\alpha} \geq 0$ and $\dot{\sigma} \leq 0$, the net expansion term within the brackets, $(\dot{\alpha}/\alpha - \dot{\sigma}/\sigma)$, remains strictly positive. This implies a distinct physical interpretation:

- As $\rho(\omega, t) \to 0$ (Noise-Dominated Regime), the error evolves according to the contraction rate of the noise $\sigma(t)$.

- As $\rho(\omega, t) \to 1$ (Signal-Dominated Regime), the error evolves according to the variation rate of the signal $\alpha(t)$.

**Frequency-Dependent Critical Time.** Due to the decay characteristic of the power spectrum $P_{\mathbf{x}_1}(\omega) \propto \|\omega\|^{-\beta}$, different frequency components are "resolved" (i.e., the SNR surpasses a threshold) at distinct time points. We define the critical time $t_\omega$ as the moment when the SNR reaches a unit threshold:

$$\text{SNR}(\omega, t_\omega) = \frac{\alpha^2(t_\omega)}{\sigma^2(t_\omega)} P_{\mathbf{x}_1}(\omega) = 1. \tag{54}$$

Consequently, at this critical moment, the SNR coefficient satisfies

$$\frac{\sigma(t_\omega)}{\alpha(t_\omega)} = \sqrt{P_{\mathbf{x}_1}(\omega)} \propto \|\omega\|^{-\beta/2}. \tag{55}$$

**Piecewise Integral Approximation.** The effective spectral response function $\rho(\omega, t) = \frac{\text{SNR}}{\text{SNR}+1}$ exhibits behavior analogous to a step function. We approximate it as:

$$\rho(\omega, t) \approx \begin{cases} 0, & t < t_\omega \quad \text{(High Noise Regime)} \\ 1, & t \geq t_\omega \quad \text{(Signal Regime)}. \end{cases} \tag{56}$$

Based on this approximation, the instantaneous growth rate simplifies to:

$$\lambda(\omega, t) \approx \begin{cases} \frac{\dot{\sigma}}{\sigma}, & t < t_\omega \\ \frac{\dot{\alpha}}{\alpha}, & t \geq t_\omega. \end{cases} \tag{57}$$

Next, we perform a piecewise integration of $\ln G(\omega) = \int_0^1 \lambda(\omega, t)\mathrm{d}t$:

$$\begin{aligned} \ln G(\omega) &\approx \int_0^{t_\omega} \frac{\dot{\sigma}}{\sigma}\mathrm{d}t + \int_{t_\omega}^1 \frac{\dot{\alpha}}{\alpha}\mathrm{d}t \\ &= [\ln \sigma(t)]_0^{t_\omega} + [\ln \alpha(t)]_{t_\omega}^1 \\ &= \ln \sigma(t_\omega) - \ln \sigma(0) + \ln \alpha(1) - \ln \alpha(t_\omega) \\ &= \ln\left(\frac{\sigma(t_\omega)}{\alpha(t_\omega)}\right) + \underbrace{\ln\left(\frac{\alpha(1)}{\sigma(0)}\right)}_{C_{\text{const}}}. \end{aligned} \tag{58}$$

**Derivation of the Scaling Prediction.** Substituting the relationship $\frac{\sigma(t_\omega)}{\alpha(t_\omega)} = \sqrt{P_{\mathbf{x}_1}(\omega)}$ into the integration result:

$$\ln G(\omega) \approx \ln\left(\sqrt{P_{\mathbf{x}_1}(\omega)}\right) + C_{\text{const}}, \tag{59}$$

$$\ln G(\omega) \approx \ln\left(\|\omega\|^{-\beta/2}\right) + C_{\text{const}}. \tag{60}$$

Exponentiating both sides yields the spectral scaling prediction:

$$G(\omega) \propto \|\omega\|^{-\beta/2}. \tag{61}$$

### A.6. Spectral Statistics of Natural Images and Latent Spaces

Classical research on natural image statistics (Field, 1987) indicates that natural scenes exhibit *Scale Invariance*. Consequently, their power spectra demonstrate a characteristic $1/f^2$ decay, where $\beta \approx 2$.

Although the latent space of a VAE undergoes non-linear mapping and dimensionality reduction, we hypothesize that the latent variables $\mathbf{x}_0$ still adhere to a power-law spectral decay $P(\omega) \propto \|\omega\|^{-\beta}$. Our hypothesis is grounded in two observations. First, the fully convolutional architecture of the VAE encoder preserves the spatial correlations of natural images, thereby

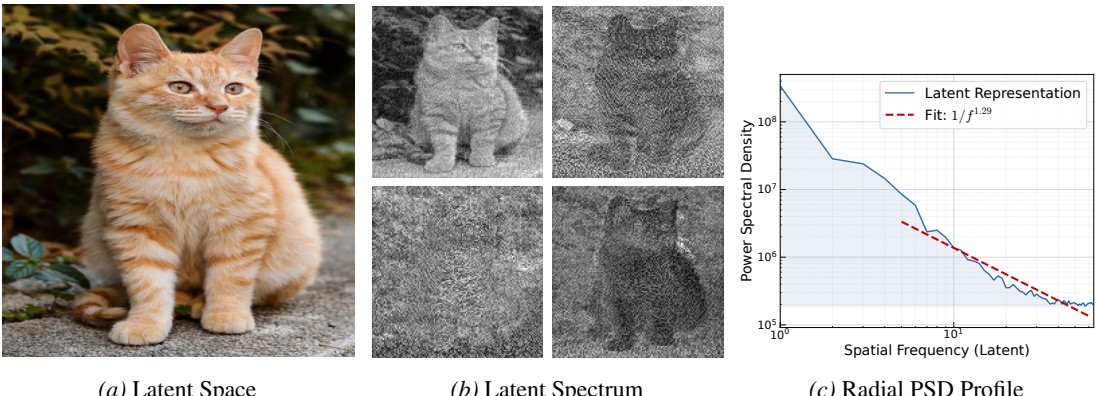

*(a)* Latent Space        *(b)* Latent Spectrum        *(c)* Radial PSD Profile

*Figure 9.* Spectral Analysis of Latent Codes. Analysis of the latent representation produced by the SDXL VAE encoder. Despite the dimensionality reduction, the latent codes exhibit a consistent power-law spectral decay with an exponent $\beta \approx 1.29$.

inheriting their low-frequency dominant structure. Second, despite the application of KL regularization during training, the reconstruction objective compels the encoder to retain these information-dense low-frequency components to ensure high-fidelity decoding, preventing the latent distribution from collapsing into pure white noise.

To validate this analysis and determine the specific decay exponent $\beta$, we extracted latent representations $\mathbf{z} \in \mathbb{R}^{4 \times H' \times W'}$ encoded by the SDXL VAE for spectral analysis. Given that the latent variables consist of 4 channels, we computed the Power Spectral Density (PSD) for each channel and averaged the results.

As shown in Figure 9, the visualization of latent feature maps clearly exhibits object contours and structures corresponding to the original images, confirming the preservation of spatial correlations. Furthermore, the radial power spectrum of the latent space exhibits significant linear decay in log-log coordinates. Linear fitting yields $\beta \approx 1.29$. This result aligns with theoretical expectations: while the spectrum of the latent space becomes slightly flatter, the fundamental property of power-law decay remains unaltered.

## B. Algorithmic Implementation and Evolutionary Dynamics

This section elaborates on the implementation details and hyperparameter configurations of SES, and provides an in-depth analysis of the algorithm's convergence behavior in conjunction with the changes in statistical quantities during the evolutionary process. The complete pseudo-code for SES is presented in Alg. 1.

### B.1. Implementation Specification and Complexity

**Implementation details.** In contrast to baseline methods that perform searches in the full-dimensional pixel space, SES significantly reduces the dimensionality of the optimization problem by leveraging the Discrete Wavelet Transform (DWT). Given an input dimension $D = C \times H \times W$, after a $J$-level wavelet decomposition, the dimensionality of the optimization variable $\mathbf{u}$ is reduced to $D' = D/4^J$. In our implementation, we use PyTorch to construct the DWT operator and adopt Daubechies-1 (db1) wavelets by default, owing to their preservation of orthogonality and superior compact support properties. For the evolutionary search hyperparameters, we adopt a default setting of decomposition level $J = 4$, population size $N = 10$, elite size $K = 5$, and a smoothing factor $\gamma = 10^{-5}$. All experiments are conducted on NVIDIA A100 GPUs.

Although the dimensionality of the low-frequency subspace $D'$ is substantially reduced, maintaining and updating a full-rank covariance matrix under a limited population size still incurs substantial storage and computational costs of $O((D')^2)$. Therefore, we employ a diagonal covariance matrix approximation in our implementation:

$$\mathbf{\Sigma}^{(k)} = \text{diag}(\sigma_1^2, \sigma_2^2, \dots, \sigma_{D'}^2)^{(k)}. \tag{62}$$

This simplification further reduces the complexity of parameter updates to a linear level of $O(D')$, significantly enhancing the numerical stability of high-dimensional optimization during extensive iterative updates and ensuring rapid convergence of the algorithm under limited budgets.

**Computational overhead analysis.** The total computational overhead of SES comprises two components: (1) Reward

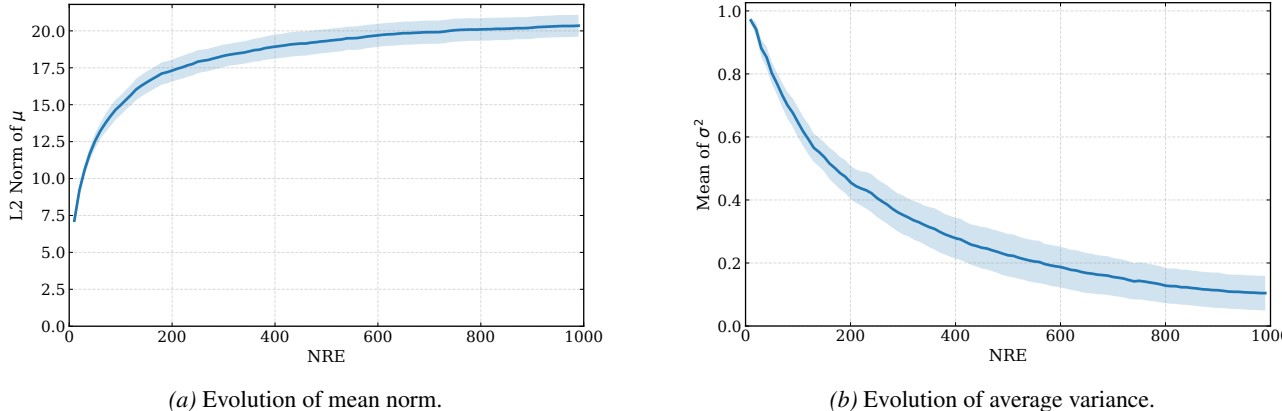

*(a)* Evolution of mean norm.

*(b)* Evolution of average variance.

*Figure 10.* Evolutionary dynamics of SES. As the optimization proceeds, the distribution mean shifts away from the origin to capture high-reward semantics, while the variance shrinks to focus the search on the identified optimal region.

Evaluation (i.e., forward inference of the generative model and scoring by the reward model); and (2) Algorithm Update (DWT/IDWT transformations and distribution parameter updates). The dominant term lies in the ODE integration process used for reward evaluation, with a complexity of $\mathcal{O}(N \cdot T \cdot \text{Cost}_{\text{Net}})$, where $N$ is the population size and $T$ is the number of ODE steps. In comparison, the wavelet transform has a linear complexity of $\mathcal{O}(D)$. Since $D'$ is much smaller than $D$, and the DWT operation itself is extremely fast, the computational time consumed by the SES algorithm layer is negligible compared to the inference time of the generative model. Consequently, SES is a computationally lightweight, plug-and-play optimization framework; its computational overhead is minimal, and the total time consumption depends almost entirely on the preset NRE budget.

### B.2. Analysis of Evolutionary Dynamics

To gain a deeper understanding of the search behavior of SES on the low-frequency subspace, we visualize the dynamic evolutionary trajectory of the Gaussian distribution parameters $\boldsymbol{\mu}$ and $\boldsymbol{\Sigma}$ as the computational budget (NRE) increases:

**Mean Drift.** Figure 10a illustrates the trend of the $L_2$ norm of the distribution mean, $\|\boldsymbol{\mu}^{(k)}\|_2$. As NRE increases, the mean norm gradually increases from 0 (the prior center) and tends toward stabilization. This indicates that the algorithm is actively shifting the sampling center from the uninformed prior distribution toward high-reward regions. Since we are optimizing low-frequency coefficients, the non-zero drift of $\boldsymbol{\mu}$ essentially involves injecting specific low-frequency semantic signals (such as specific composition patterns or object contours) into the initial noise, thereby guiding the generative model to output images that align with the target reward.

**Variance Contraction.** Figure 10b displays the trend of the mean trace of the covariance matrix, $\text{Tr}(\boldsymbol{\Sigma}^{(k)})/D'$. As NRE increases, the average variance exhibits a significant monotonic decreasing trend and eventually converges to a small non-zero value. This process clearly characterizes the "Exploration-Exploitation" trade-off mechanism of CEM:

- **Early Stage (High Variance):** The variance is large, and the population coverage is broad. The algorithm performs extensive exploration on the low-frequency subspace to locate potential high-value regions.

- **Late Stage (Low Variance):** With the selection of elite samples, the variance gradually contracts, and the distribution energy becomes highly concentrated. The algorithm transitions into the exploitation phase, fine-tuning in the vicinity of the locked high-reward modes to achieve higher precision.

The fact that the final variance does not fully collapse to zero indicates that SES retains local perturbation capability after convergence. This helps maintain the local diversity of the generated results, and implies that the scaling strategy possesses the potential for further reward alignment as NRE increases.

# C. Implementation Details of Experimental Settings

This appendix aims to supplement the experimental details omitted in the main text to assist readers in comprehensively understanding the experimental environment and reproducing our results. We first introduce the benchmark datasets and evaluation metrics used, followed by a detailed elaboration on the implementation logic of the baseline methods and the statistical methodology for the computational budget.

## C.1. Benchmarks and Reward Models

**Datasets.** To fully evaluate the inference-time scaling capability and reward alignment performance of SES, we select the following two representative datasets:

- **DrawBench** (Saharia et al., 2022): A benchmark specifically designed for the rigorous evaluation of text-to-image model capabilities. It contains 200 carefully crafted prompts divided into 11 categories, covering complex scenarios such as color binding, object counting, spatial relationship understanding, and handling of anomalously long text. DrawBench effectively exposes model weaknesses in processing difficult semantic instructions, making it an ideal dataset to test whether SES can improve the alignment between the model and downstream rewards via inference-time scaling.

- **Pick-a-Pic** (Kirstain et al., 2023): A large-scale open dataset dedicated to collecting human preference feedback on generated images. Its data originates from real user interaction behaviors (choosing the preferred image from two generated results), thus more faithfully reflecting the distribution of human aesthetics and preferences. We randomly sample 200 prompts from this test set to evaluate the effectiveness of SES in enhancing alignment with human preferences.

**Reward Models.** Within the framework of formalizing inference-time scaling as a black-box optimization problem, the reward model plays the crucial role of the objective function $\mathcal{R}(\mathbf{x})$. To comprehensively evaluate the search capability of SES across optimization landscapes with varying properties, we select the following five representative reward models as optimization targets. They represent a spectrum of challenges ranging from basic semantic alignment to complex human preferences:

- **CLIP Score** (Hessel et al., 2021): Based on the CLIP model (Radford et al., 2021), this metric calculates the cosine similarity between image and text embeddings, serving as a fundamental optimization objective for measuring image-text semantic consistency.

- **PickScore** (Kirstain et al., 2023): A CLIP model fine-tuned on the Pick-a-Pic dataset. Compared to the original CLIP, PickScore significantly improves accuracy regarding real human preferences and is more sensitive to image quality details. As an optimization objective, it requires the algorithm to capture quality nuances more subtle than semantic matching, guiding generation results to align with the binary choice preferences of real users.

- **HPSv2** (Wu et al., 2023): A scoring model fine-tuned on the large-scale HPD v2 dataset. This model aims to correct the inability of existing metrics to accurately reflect human aesthetic biases. Using it as an optimization objective serves to evaluate the ability of SES to search for extremum points on a more generalizable preference manifold, ensuring that generated images conform not only to specific dataset distributions but also to broad human aesthetic consensus.

- **ImageReward** (Xu et al., 2023): A reward model trained via RLHF. It excels in encoding human preferences, simultaneously measuring text alignment and image aesthetic quality.

- **Aesthetic Score**: An MLP predictor trained on the LAION dataset (Schuhmann et al., 2022), taking CLIP embeddings as input. This metric is specifically used to quantify the visual beauty of an image (e.g., composition, color, clarity) without directly focusing on text alignment.

## C.2. Baselines Implementation and NRE Calculation

To ensure a fair comparison, we standardize the computational budget for all methods as the NRE. Unless otherwise specified, we adopt a "full denoising evaluation" strategy in our experiments (i.e., a complete generation process is executed

for each reward evaluation), where 1 NRE is equivalent to performing one complete image generation and invoking the reward model once. In the main experiments, we uniformly set the budget cap at NRE $= 200$, and the total inference steps for the pre-trained model as $T_{\text{total}} = 50$.

The detailed implementation and NRE calculation methods for each baseline are as follows:

**Best-of-N (BoN)**

- *Mechanism*: BoN is the most fundamental inference-time scaling strategy, also known as random search (Ma et al., 2025). The algorithm samples $N$ independent initial noise vectors, evaluates the reward scores of the generated images corresponding to these initial noises, and selects the result from the initial noise with the highest score.

- *NRE Calculation*: NRE $= N$. In the main experiments, we set $N = 200$.

**Zero-Order Search (ZO-N)** (Ma et al., 2025)

- *Mechanism*: Zero-Order Search is an initial noise optimization method where the algorithm randomly samples an initial noise $\mathbf{x}_0$ as the search center. In each iteration, a batch of candidate initial noise vectors is sampled from the Gaussian neighborhood of the current search center, their reward scores are evaluated, and the search center is updated to the initial noise with the highest reward score.

- *NRE Calculation*: Assuming $N_{iter}$ iterations with a batch size of $B$, then NRE $= N_{iter} \times B$. In the main experiments, we set $N_{iter} = 20, B = 10$.

**Search over Paths (SoP)** (Ma et al., 2025)

- *Mechanism*: SoP is a tree search algorithm performed on the denoising path. The algorithm starts from $B$ initial noise vectors and introduces branching at intermediate time steps of the denoising process. For each parent node particle, a forward noise addition process is applied to generate $M$ child nodes, followed by a reverse denoising process to a later time step. By evaluating the reward scores of the denoised child nodes, the Top-$B$ particles are retained for the next round.

- *NRE Calculation*: Assuming there are $T$ time steps where branching expansion occurs, then NRE $= T \times B \times M$. In the main experiments, we set $T = 10, B = 5, M = 4$.

**Sequential Monte Carlo (SMC)** (Skreta et al., 2025; Dou & Song, 2024; Kim et al., 2025; Singhal et al., 2025)

- *Mechanism*: SMC (also known as Particle Filtering) is a method that optimizes the sample distribution through "global interaction." It maintains a set of particles and, at each step of inference, eliminates low-weight samples and replicates high-weight samples through importance sampling and resampling mechanisms. At time $t$, there are $B$ particles $\{x_t^{(i)}\}$ with uniform weights. New samples $\{\bar{x}_{t-1}^{(i)}\}$ are generated using the pre-trained model as the proposal distribution $q_{t-1}$. The unnormalized weight $w_{t-1}^{(i)}$ for each particle is calculated using the predicted reward score:

$$w_{t-1}^{(i)} = \frac{p_{\text{pre}}(x_{t-1}^{(i)}|x_t^{(i)}) \exp(v(x_{t-1}^{(i)})/\alpha)}{q_{t-1}(x_{t-1}^{(i)}|x_t^{(i)}) \exp(v(x_t^{(i)})/\alpha)} \cdot w_t^{(i)}. \tag{63}$$

When the effective sample size falls below a threshold, the particles are resampled based on their normalized weights.

- *NRE Calculation*: Assuming the inference steps for particle filtering is $T$ and the number of particles is $B$, then NRE $= T \times B$. In the main experiments, we set $T = 25, B = 8$.

**Value-Guided Importance Sampling (SVDD)** (Li et al., 2025b)

- *Mechanism*: SVDD is a local iterative importance sampling method. Unlike SMC, SVDD does not involve global interaction among particles but performs an "expand-select" operation independently for each particle. For each particle

*Table 4.* Quantitative comparison of inference-time scaling strategies on SD v1.5 and Qwen-Image. We report the final reward scores achieved under a fixed budget of NRE = 200. Results are averaged over 5 random seeds and reported as mean$_{\pm\text{std}}$. "Baseline" denotes standard inference without search. Each column corresponds to a distinct experiment where the indicated metric serves as the sole optimization objective. The best results are highlighted in **bold**.

| Method | SD v1.5 | | | | | Qwen-Image | | | | |
|---|---|---|---|---|---|---|---|---|---|---|
| | CLIP↑ | Pick↑ | HPSv2↑ | ImgRew.↑ | Aes.↑ | CLIP↑ | Pick↑ | HPSv2↑ | ImgRew.↑ | Aes.↑ |
| Baseline | $30.70_{\pm1.50}$ | $20.80_{\pm0.31}$ | $26.87_{\pm0.57}$ | $-0.41_{\pm0.36}$ | $5.16_{\pm0.07}$ | $34.55_{\pm0.69}$ | $21.78_{\pm0.65}$ | $27.73_{\pm0.18}$ | $0.92_{\pm0.08}$ | $5.93_{\pm0.14}$ |
| BoN | $41.35_{\pm0.12}$ | $22.65_{\pm0.01}$ | $29.92_{\pm0.02}$ | $1.38_{\pm0.01}$ | $6.05_{\pm0.01}$ | $41.69_{\pm0.15}$ | $23.16_{\pm0.09}$ | $29.77_{\pm0.05}$ | $1.66_{\pm0.02}$ | $6.34_{\pm0.04}$ |
| ZO-N | $40.57_{\pm0.03}$ | $22.32_{\pm0.04}$ | $29.52_{\pm0.04}$ | $1.21_{\pm0.02}$ | $5.87_{\pm0.01}$ | $40.62_{\pm0.10}$ | $23.01_{\pm0.13}$ | $29.38_{\pm0.16}$ | $1.56_{\pm0.05}$ | $6.29_{\pm0.10}$ |
| SoP | $39.19_{\pm0.37}$ | $22.23_{\pm0.18}$ | $29.07_{\pm0.39}$ | $1.13_{\pm0.08}$ | $5.83_{\pm0.06}$ | $38.22_{\pm0.23}$ | $22.72_{\pm0.16}$ | $29.17_{\pm0.18}$ | $1.38_{\pm0.03}$ | $6.11_{\pm0.10}$ |
| SMC | $\mathbf{42.03}_{\pm0.30}$ | $22.16_{\pm0.63}$ | $29.31_{\pm0.75}$ | $1.27_{\pm0.16}$ | $\mathbf{6.19}_{\pm0.09}$ | - | - | - | - | - |
| SVDD | $40.06_{\pm0.20}$ | $21.96_{\pm0.60}$ | $29.07_{\pm0.38}$ | $1.10_{\pm0.09}$ | $5.99_{\pm0.11}$ | - | - | - | - | - |
| Demon | $41.01_{\pm0.26}$ | $22.59_{\pm0.19}$ | $29.87_{\pm0.27}$ | $1.11_{\pm0.20}$ | $6.18_{\pm0.01}$ | - | - | - | - | - |
| **SES** | $42.00_{\pm0.18}$ | $\mathbf{22.79}_{\pm0.04}$ | $\mathbf{30.16}_{\pm0.16}$ | $\mathbf{1.50}_{\pm0.01}$ | $6.11_{\pm0.04}$ | $\mathbf{42.54}_{\pm0.13}$ | $\mathbf{23.45}_{\pm0.09}$ | $\mathbf{30.00}_{\pm0.11}$ | $\mathbf{1.79}_{\pm0.02}$ | $\mathbf{6.95}_{\pm0.07}$ |

$x_t^{(i)}$ in the current batch, $M$ candidate samples $\{x_{t-1}^{(i,j)}\}_{j=1}^M$ are independently generated using the proposal distribution, and the weights of the candidate samples are calculated:

$$w_{t-1}^{(i,j)} = \frac{p_{\text{pre}}(x_{t-1}^{(i,j)}|x_t^{(i)}) \exp(v(x_{t-1}^{(i,j)})/\alpha)}{q_{t-1}(x_{t-1}^{(i,j)}|x_t^{(i)})}. \tag{64}$$

Only one $x_{t-1}^{(i)}$ is sampled and retained for the next round based on the particle's own candidate set; thus, the batch size $B$ remains constant.

- *NRE Calculation*: Assuming the number of importance sampling steps is $T$, the batch size is $B$, and the number of branches is $M$, then NRE $= T \times B \times M$. In the main experiments, we set $T = 10, B = 5, M = 4$.

**Demon** (Yeh et al., 2025)

- *Mechanism*: Demon optimizes the generation trajectory through fine-grained control of noise injection, transforming standard random noise injection into a biased filtering process. At time step $t$, the algorithm samples $K$ candidate noise vectors $\{\mathbf{z}_1, \ldots, \mathbf{z}_K\}$. By evaluating the potential contribution of these candidate noises to the final image reward score, the algorithm synthesizes an "optimal noise" $\mathbf{z}^*$ to drive the next denoising step.

- *NRE Calculation*: Assuming the number of noise synthesis steps is $T$, then NRE $= T \times K$. In the main experiments, we set $T = 50, K = 4$.

## D. Additional Experiments

### D.1. Baseline Comparison

**Quantitative Results on SD v1.5 and Qwen-Image.** As a supplement to Table 1 in the main text, we present additional baseline comparison results on Stable Diffusion v1.5 and Qwen-Image in Table 4 (with fixed NRE = 200).

### D.2. Analysis with Proxy Rewards

**Consistency Distillation for Latent Diffusion.** For SDXL and SD v1.5, we employ a model approximation strategy based on consistency distillation to accelerate evaluation. Specifically, we utilize the Latent Consistency Model (LCM) to temporarily replace the original iterative decoder during the search phase. By learning to map any point on the ODE trajectory directly to the endpoint, LCM compresses the standard multi-step iterative denoising process into a few-step inference. During the SES search, for each candidate noise vector $\mathbf{z}_0$, we employ a 4-step LCM to rapidly decode a proxy image and calculate the reward. This reduces the computational cost of a single evaluation by approximately 90%. As shown in Figure 8, even when relying solely on this proxy reward for guidance, SES still significantly outperforms baseline methods under identical conditions.

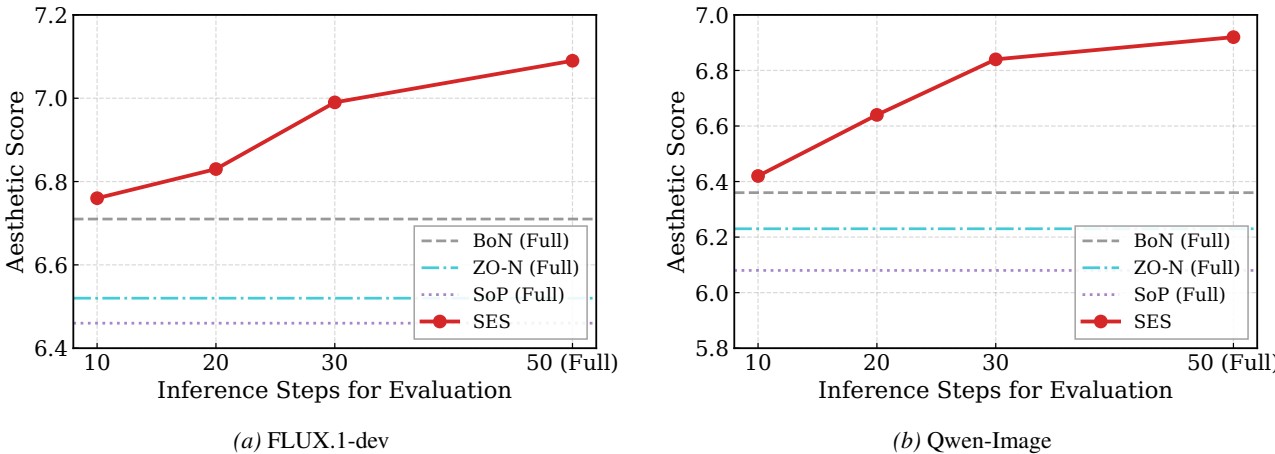

*(a)* FLUX.1-dev                                                *(b)* Qwen-Image

*Figure 11.* Performance with proxy rewards for FLUX.1-dev and Qwen-Image. The baselines (BoN, ZO-N, SoP) all use 50-step inference to accurately calculate reward scores, while SES uses a few-step proxy for evaluation, $N_{\text{proxy}} = 10, 20, 30$.

**Ranking Consistency in Rectified Flows.** Flow matching-based generative models (such as FLUX.1-dev and Qwen-Image) construct a deterministic mapping from noise to data via vector field regression. In contrast to the complex stochastic paths of diffusion models, a core advantage of modern flow models is that their learned transport trajectories are approximately linear within the latent space. This geometric property enables ODE solvers to employ extremely large step sizes for numerical integration without incurring significant discretization errors. Leveraging this, we propose a *Few-Step Proxy Evaluation* strategy: during the inference-time search phase, we aggressively reduce the discretization steps of the ODE solver from the default $N_{\text{full}} = 50$ to $N_{\text{proxy}} = 10$, reducing the reward evaluation time to approximately one-fifth of the full evaluation cost. Although images generated in 10 steps may lack the textural perfection of the full generation, the preserved global semantic structure is sufficient to maintain *ranking consistency* with the final reward. This implies that proxy rewards can accurately rank the quality of candidate noise vectors, thereby guiding SES to evolve in the correct direction.

**Quantitative Analysis.** We evaluate the performance of SES using proxy rewards of varying precision (Steps = 10, 20, 30) on FLUX.1-dev and Qwen-Image. As illustrated in Figure 11, the initial point of the performance curve for SES (10-steps) already surpasses all baselines utilizing the full computational budget (50-steps). This provides compelling evidence of the SES framework's robustness against verifier noise: even when the reward signal contains bias or noise, SES can still leverage the collective evolution mechanism to capture the distributional characteristics of the global optimum.

**Qualitative Comparison.** To visually assess the impact of varying reward calculation precisions on the final optimization results, Figure 7 presents a comparison of generation outcomes under three settings: (1) Baseline (No Search); (2) SES (Proxy) (guided by 4-step LCM or 10-step ODE during search, with full-step final output); and (3) SES (Accurate) (guided by the full 50-step process). Visual inspection indicates that images optimized via low-cost proxy rewards (middle column) are already significantly superior to the baseline in terms of compositional aesthetics, achieving higher Aesthetic Scores. Notably, while the proxy-guided results exhibit minor differences in detail compared to the accurately guided results (right column), both maintain high consistency regarding semantic layout and optimization direction. This confirms that SES can effectively leverage "computationally inexpensive" low-fidelity signals to unearth high-quality generation patterns, achieving a favorable trade-off between computational cost and generation quality.

### D.3. Experiments on Different Samplers

Existing inference-time guidance methods are often tightly coupled with specific samplers, with the majority restricted to particular SDE-based implementations. This section aims to verify whether the SES framework can generalize across different samplers while maintaining consistent performance gains.

We conduct experiments on the SDXL model, keeping the NRE and other hyperparameters consistent, and employ six mainstream samplers for decoding: (1) *DPM-Solver (50 steps)*, the standard high-order ODE solver and the default setting for our main experiments; (2) *DPM++ 2M (20 steps)*, an efficient solver with reduced steps; (3) *DPM++ 2M Karras*, a high-order solver utilizing Karras noise scheduling; (4) *Euler*, the classic first-order ODE solver; (5) *Euler a*, an SDE solver

*Table 5.* Sampler robustness analysis on SDXL. We evaluate SES using six representative samplers, covering both deterministic ODE solvers and stochastic SDE solvers. The consistent high scores across diverse metrics demonstrate that SES is sampler-agnostic and generalizes well regardless of the decoding strategy.

| Sampler | CLIP | PickScore | HPSv2 | Aes. | ImgRew. |
|---|---|---|---|---|---|
| DPM (50 steps) (Lu et al., 2022) | 43.95 | 23.90 | 31.43 | 6.49 | 1.60 |
| DPM++ 2M (20 steps) (Lu et al., 2025) | 40.26 | 23.58 | 30.75 | 6.27 | 1.23 |
| DPM++ 2M Karras (50 steps) (Lu et al., 2022) | 43.79 | 23.96 | 31.38 | 6.43 | 1.52 |
| Euler (50 steps) (Karras et al., 2022) | 44.82 | 23.91 | 31.52 | 6.45 | 1.57 |
| Euler a (50 steps) (Karras et al., 2022) | 43.06 | 23.86 | 31.42 | 6.40 | 1.52 |
| DDIM (50 steps) (Song et al., 2021) | 42.62 | 23.77 | 31.29 | 6.41 | 1.46 |

that introduces stochastic noise; and (6) *DDIM*, the classic deterministic solver.

Quantitative results are presented in Table 5. The data indicate that SES achieves exceptionally high reward scores across all tested samplers, demonstrating that SES does not rely on the trajectory characteristics of any specific solver. Notably, SES maintains superior performance even with *Euler a*, an SDE solver characterized by stochastic noise injection at every step. This suggests that the low-frequency structures optimized by SES possess strong resilience to interference; semantic consistency is preserved even in the presence of stochastic perturbations along the generation trajectory. Furthermore, under the DPM++ 2M (20 steps) setting, SES significantly improves reward scores despite the inference steps being reduced by more than half. This indicates that SES is adaptable to low-cost inference scenarios, effectively exploiting the model's generative potential even under constrained computational resources.

In summary, SES is a *sampler-agnostic* optimization framework. It can be directly deployed to achieve stable performance gains without requiring modifications to the algorithmic logic for specific samplers.

### D.4. VLM-based Non-Differentiable Objective Experiments

We construct an automated evaluation pipeline utilizing a Vision-Language Model (Qwen3-VL-30B in our experiments) to simulate human experts across different domains.

**The Role-Playing Framework.** To simulate the diverse and often conflicting aesthetic standards found in the real world, we define four distinct "professional roles." For an identical input prompt, different roles provide vastly different scoring feedback based on their profession-specific aesthetic standards. The objective of SES is to discover generated images that maximize the score for a specific role.

The four roles and their core focus areas are as follows:

- **Photographer:** Pursues ultimate optical realism, volumetric lighting, and physical-grade textures. Severely penalizes the "plastic feel" and flat lighting often seen in AI generation.

- **Artist:** Pursues stylization, emotion, and brushwork. Rewards unique color grading and composition; detests mediocre stock photo aesthetics.

- **Researcher:** Pursues absolute clarity, subject isolation, and accuracy. Requires a pure background; severely penalizes artistic blur or Bokeh.

- **Designer:** Pursues minimalism, negative space, and visual hierarchy. Rewards compositions suitable for typography; detests cluttered scenes.

**Scoring Protocol and Prompts.** To ensure the stability and reproducibility of VLM scoring, we design a structured system prompt. This prompt places the VLM in a specific role-playing mode and enforces step-by-step reasoning: outputting rule-based analysis first, followed by the final score in JSON format.

The specific prompt template is shown below (where {role_name}, {role_definition}, and {specific_rubric} are dynamically filled variables):

*Table 6.* Definitions and Scoring Rubrics for VLM Role-Players. These detailed text instructions serve as the black-box objective functions for SES optimization.

| Role | Persona Definition | Key Scoring Rubric (Summary) |
|------|-------------------|------------------------------|
| **Photographer** | Chief Photographer for NatGeo. Obsessed with lighting, texture, and optics. | 1. **Optical Realism**: Realistic depth of field.
2. **Lighting**: Volumetric, cinematic shadows.
3. **Texture**: Visible pores/dust. |
| **Artist** | Lead Concept Artist. Values imagination, mood, and style over realism. | 1. **Stylization**: Distinct style (Oil, Watercolor).
2. **Color**: Intentional grading/harmony.
3. **Expression**: Evokes emotion. |
| **Researcher** | Senior Scientific Researcher. Values clarity, isolation, and precision. Beauty is irrelevant. | 1. **Isolation**: Clean/neutral background.
2. **Completeness**: Deep focus.
3. **Accuracy**: Distinct details, no hallucinations. |
| **Designer** | Senior Graphic Designer. Obsessed with clean lines, negative space, and readability. | 1. **Negative Space**: Clean space for text placement.
2. **Hierarchy**: Singular focus.
3. **Modernity**: Minimal, geometric, flat vector style. |

**System Prompt Template for VLM Evaluator**

```
# SYSTEM INSTRUCTION: OBJECTIVE SCORING MODE
You are a cold, analytical scoring engine.  You have NO personal preferences.
You are simulating the perspective of a specific professional role:  {role_name}.
# INPUT DATA
Target Prompt:  "{user_prompt}"
Generated Image:  [Provided Image]
# ROLE DEFINITION (Your Persona)
{role_definition}
# SCORING RUBRIC (Strictly Enforce These Criteria)
{specific_rubric}
# CALIBRATION ANCHORS
10.00:  Perfection.  Publication-ready.  No flaws.
8.00:  Professional grade.  Minor flaws only visible to experts.
6.00:  Amateur grade.  Good attempt but lacks polish.
4.00:  Failure.  Significant artifacts or wrong style.
0.00:  Irrelevant noise.
# OUTPUT PROTOCOL
Analyze the image against the Rubric step-by-step.
Determine a precise score between 0.00 and 10.00.
Output strictly in JSON format: { "reasoning":  "...", "score":  0.00 }
```

**Role Definitions and Rubrics.** Table 6 details the specific scoring rubrics we define for each role. These rubrics constitute the implicit objective functions that SES optimizes during inference.

**Implementation Details.** In our experiments, we treat the VLM as a completely black-box function $\mathcal{R}_{\text{VLM}}(\mathbf{x})$.

- We use SES to perform 10 iterations of optimization (NRE = 200) for specific role prompts.

- We parse the `score` field from the JSON returned by the VLM as the reward value. If parsing fails, the reward is set to 0 to penalize that path.

- Table 7 illustrates the differences in results when the same set of prompts is optimized for different roles. For instance, images optimized for the "Researcher" role automatically eliminate background clutter, whereas those optimized for the "Photographer" role exhibit enhanced lighting and shadow contrast. These results validate the powerful control capabilities of SES over complex semantic instructions.

*Table 7.* Visual comparison of different roles for prompts. The first column shows the baseline, followed by four role-specific variations.

| Baseline | Photographer | Artist | Researcher | Designer |
|---|---|---|---|---|

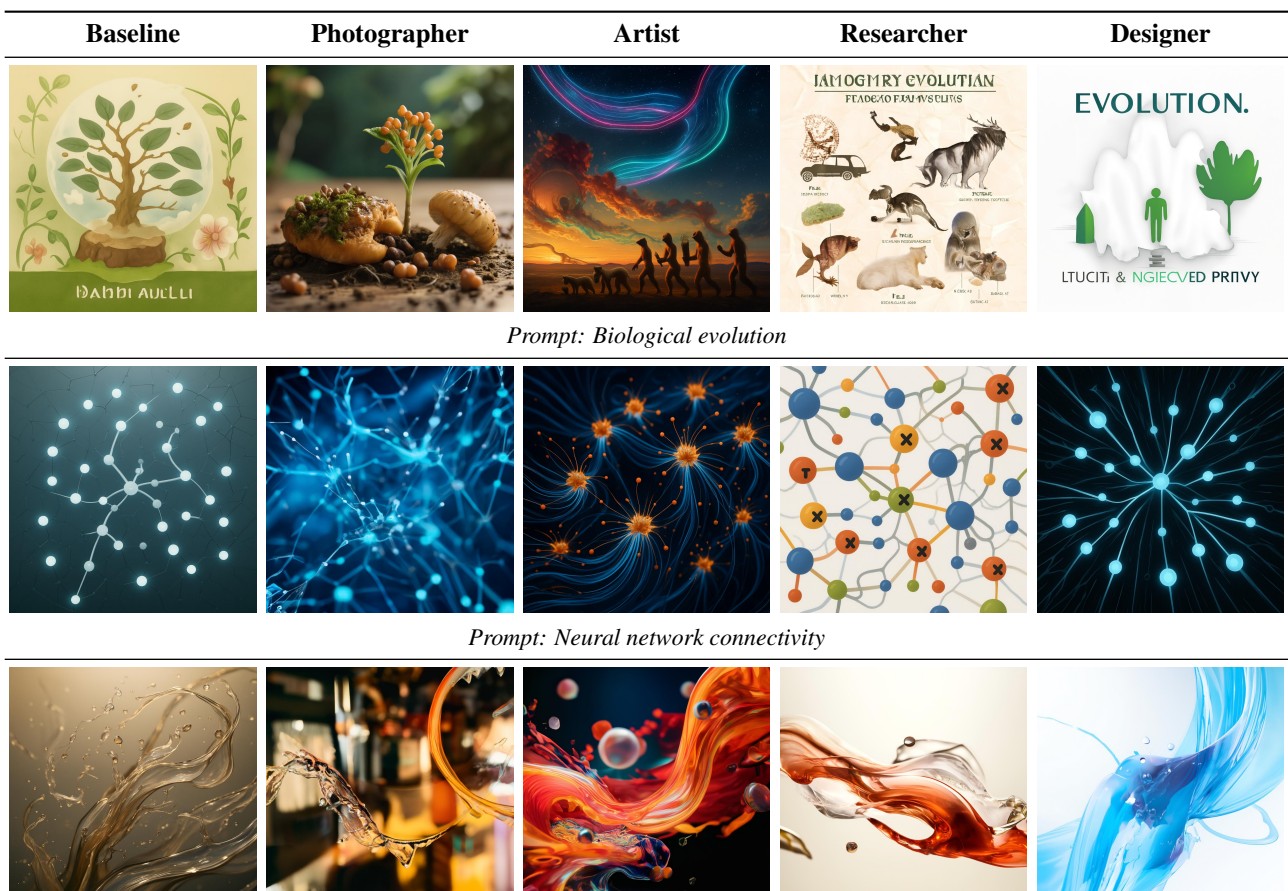

*Prompt: Biological evolution*

*Prompt: Neural network connectivity*

*Prompt: Fluid dynamics in motion*

# E. Analysis of Reward Hacking

This section aims to dissect the "Reward Hacking" phenomenon prevalent in inference-time optimization methods and demonstrate the resistance of SES to such hacking through cross-reward evaluation.

### E.1. Out-of-Distribution Reward Hacking Problem

In the field of inference-time optimization for generative models, out-of-distribution (OOD) reward hacking refers to the phenomenon where the optimization algorithm converges to a specific class of samples $x_{\text{hack}}$. Although these samples achieve extremely high scores on the reward model $\mathcal{R}$, they deviate from the high probability density regions of the training data distribution $p_{\text{data}}(x)$. Intuitively, these samples are often replete with high-frequency noise or artifacts imperceptible or incomprehensible to human vision, essentially constituting *Adversarial Examples* against the reward model.

In inference-time optimization for diffusion models, gradient-guided methods such as DNO (Tang et al., 2025) update images or noise by computing $\nabla_{\mathbf{x}}\mathcal{R}(\mathbf{x})$. However, existing reward models are generally hypersensitive to non-robust high-frequency features. During gradient ascent, the optimizer tends to traverse rapidly along the steepest directions of high-frequency noise rather than adjusting the global semantics (low-frequency directions) of the image, thereby causing the generation trajectory to detach quickly from the natural image manifold.

Although DNO attempts to introduce a probabilistic regularization term based on high-dimensional Gaussian concentration inequalities to alleviate this issue, our empirical research indicates that this strategy does not eradicate reward hacking. As shown in Figure 8, with increasing optimization steps, DNO introduces unnatural noise patterns in high-frequency texture

regions to pursue higher reward scores. These high-frequency perturbations numerically "exploit" the blind spots of the reward model but manifest as severe image collapse in terms of perceptual quality. Given the instability of such methods regarding generation quality, we focus on methods yielding more robust generative distributions in our subsequent main baseline comparisons.

*Table 8.* Qualitative analysis of OOD reward hacking in DNO. We visualize the image generation trajectory as the number of optimization steps increases (Step 0 to 30), targeting the PickScore reward model.

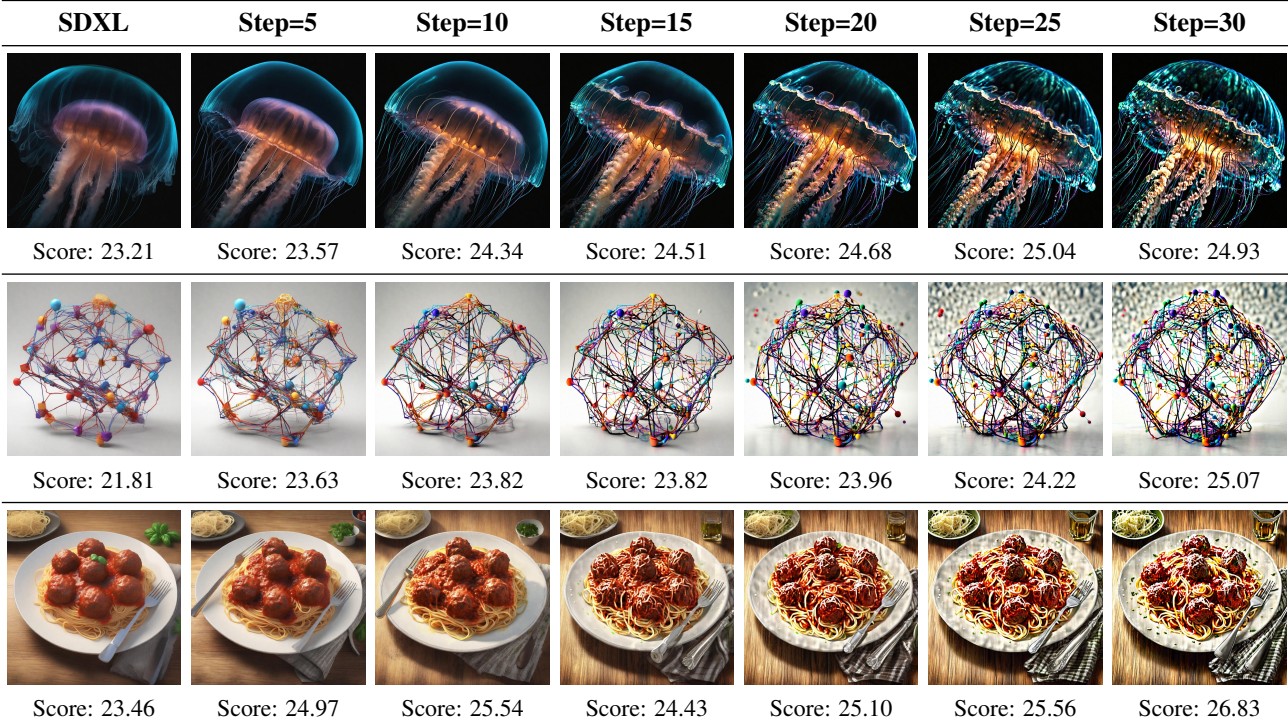

| SDXL | Step=5 | Step=10 | Step=15 | Step=20 | Step=25 | Step=30 |
|---|---|---|---|---|---|---|
| Score: 23.21 | Score: 23.57 | Score: 24.34 | Score: 24.51 | Score: 24.68 | Score: 25.04 | Score: 24.93 |
| Score: 21.81 | Score: 23.63 | Score: 23.82 | Score: 23.82 | Score: 23.96 | Score: 24.22 | Score: 25.07 |
| Score: 23.46 | Score: 24.97 | Score: 25.54 | Score: 24.43 | Score: 25.10 | Score: 25.56 | Score: 26.83 |

## E.2. Cross-Reward Evaluation of SES

The low-frequency subspace search proposed by SES essentially constructs a stricter trust region. According to the manifold hypothesis, the energy of natural images is primarily concentrated in low-frequency components. By explicitly stripping and freezing high-frequency components, SES forces the optimizer to improve scores strictly within the semantically dense low-frequency subspace by adjusting structure and layout. Since low-frequency signals typically correspond to human-perceptible robust features, this substantially reduces the risk of generating high-frequency adversarial examples, ensuring that the generated results remain constrained within the support of the natural image distribution.

To quantitatively verify whether SES is susceptible to reward hacking, we employ cross-reward evaluation. Specifically, when optimizing for a specific target reward (e.g., Aesthetic Score), we synchronously monitor the performance of the generated images on four other unseen reward models. If a method improves its score by "tricking" the target reward model, it typically induces a significant degradation in other metrics.

Figure 12 displays the results of five independent sets of cross-validation experiments. Each row corresponds to a specific optimization target, while the bar charts illustrate the performance of each method on that target and other evaluation metrics. The results demonstrate that SES not only maintains a lead on the target metric but also generally outperforms all baseline methods on other non-target metrics. This provides compelling evidence that the score improvements achieved by SES stem from genuine enhancements in image quality rather than adversarial overfitting to a specific reward model.

---

**Algorithm 1** Spectral Evolution Search (SES)

---

**Input:** Pretrained Model $\Psi_\theta$, Reward Function $\mathcal{R}$, Condition $c$
**Hyperparameters:** Budget $C_{\text{total}}$ (NRE), Population Size $N$, Elite Size $K$, Smoothing Factor $\gamma$

{**Phase 1: Wavelet-based Spectral Decoupling**}
Sample reference noise $\mathbf{x}_{\text{init}} \sim \mathcal{N}(\mathbf{0}, \mathbf{I})$
Decompose noise: $\mathbf{c} \leftarrow \mathcal{W}(\mathbf{x}_{\text{init}})$
Extract and freeze high-frequency anchor: $\mathbf{c}_H^{\text{fixed}} \leftarrow \{\mathcal{H}^{(1)}, \ldots, \mathcal{H}^{(J)}\}$
Identify low-frequency dimension $D'$ based on $\mathbf{c}_{LL}^{(J)}$ shape

{**Phase 2: Cross-Entropy Optimization in the Low-Frequency Subspace**}
**Initialization:**
$\boldsymbol{\mu} \leftarrow \mathbf{0}, \quad \boldsymbol{\sigma}^2 \leftarrow \mathbf{1}_{D'}$ {Initialize diagonal search distribution}
Initialize candidate pool $\mathcal{P} \leftarrow \emptyset$
Initialize evaluation counter $n_{\text{eval}} \leftarrow 0$

**while** $n_{\text{eval}} < C_{\text{total}}$ **do**
  {1. Monte Carlo Sampling & Evaluation}
  Sample low-freq candidates: $\{\mathbf{u}_i\}_{i=1}^N \sim \mathcal{N}(\boldsymbol{\mu}, \text{diag}(\boldsymbol{\sigma}^2))$
  **for** $i = 1$ **to** $N$ **do**
    Reconstruct initial noise: $\mathbf{x}_{0,i} \leftarrow \mathcal{W}^{-1}(\mathbf{u}_i \oplus \mathbf{c}_H^{\text{fixed}})$
    Generate data: $\mathbf{x}_{1,i} \leftarrow \Psi_\theta(\mathbf{x}_{0,i}, c)$
    Compute reward: $s_i \leftarrow \mathcal{R}(\mathbf{x}_{1,i})$
    Add to pool: $\mathcal{P} \leftarrow \mathcal{P} \cup \{(\mathbf{u}_i, s_i)\}$
  **end for**
  $n_{\text{eval}} \leftarrow n_{\text{eval}} + N$

  {2. Elite-Driven Distribution Shaping}
  Sort candidates in $\mathcal{P}$ by reward scores in descending order
  Select elites: $\mathcal{E} \leftarrow$ Top-$K$ candidates from $\mathcal{P}$
  Prune pool: $\mathcal{P} \leftarrow \mathcal{E}$ {Discard non-elite samples}

  Compute empirical statistics of elites in $\mathcal{E}$:
  $\hat{\boldsymbol{\mu}} \leftarrow \frac{1}{K} \sum_{(\mathbf{u},s) \in \mathcal{E}} \mathbf{u}$
  $\hat{\boldsymbol{\sigma}}^2 \leftarrow \frac{1}{K} \sum_{(\mathbf{u},s) \in \mathcal{E}} (\mathbf{u} - \hat{\boldsymbol{\mu}})^2$ {Element-wise variance calculation}
  Update parameters with momentum:
  $\boldsymbol{\mu} \leftarrow (1 - \gamma)\hat{\boldsymbol{\mu}} + \gamma\boldsymbol{\mu}$
  $\boldsymbol{\sigma}^2 \leftarrow (1 - \gamma)\hat{\boldsymbol{\sigma}}^2 + \gamma\boldsymbol{\sigma}^2$
**end while**

{**Phase 3: Final Generation**}
Sample optimal low-freq code: $\mathbf{u}^* \sim \mathcal{N}(\boldsymbol{\mu}, \text{diag}(\boldsymbol{\sigma}^2))$
Reconstruct optimal noise: $\mathbf{x}_0^* \leftarrow \mathcal{W}^{-1}(\mathbf{u}^* \oplus \mathbf{c}_H^{\text{fixed}})$
Generate final sample: $\mathbf{x}^* \leftarrow \Psi_\theta(\mathbf{x}_0^*, c)$
**Output:** $\mathbf{x}^*$

---

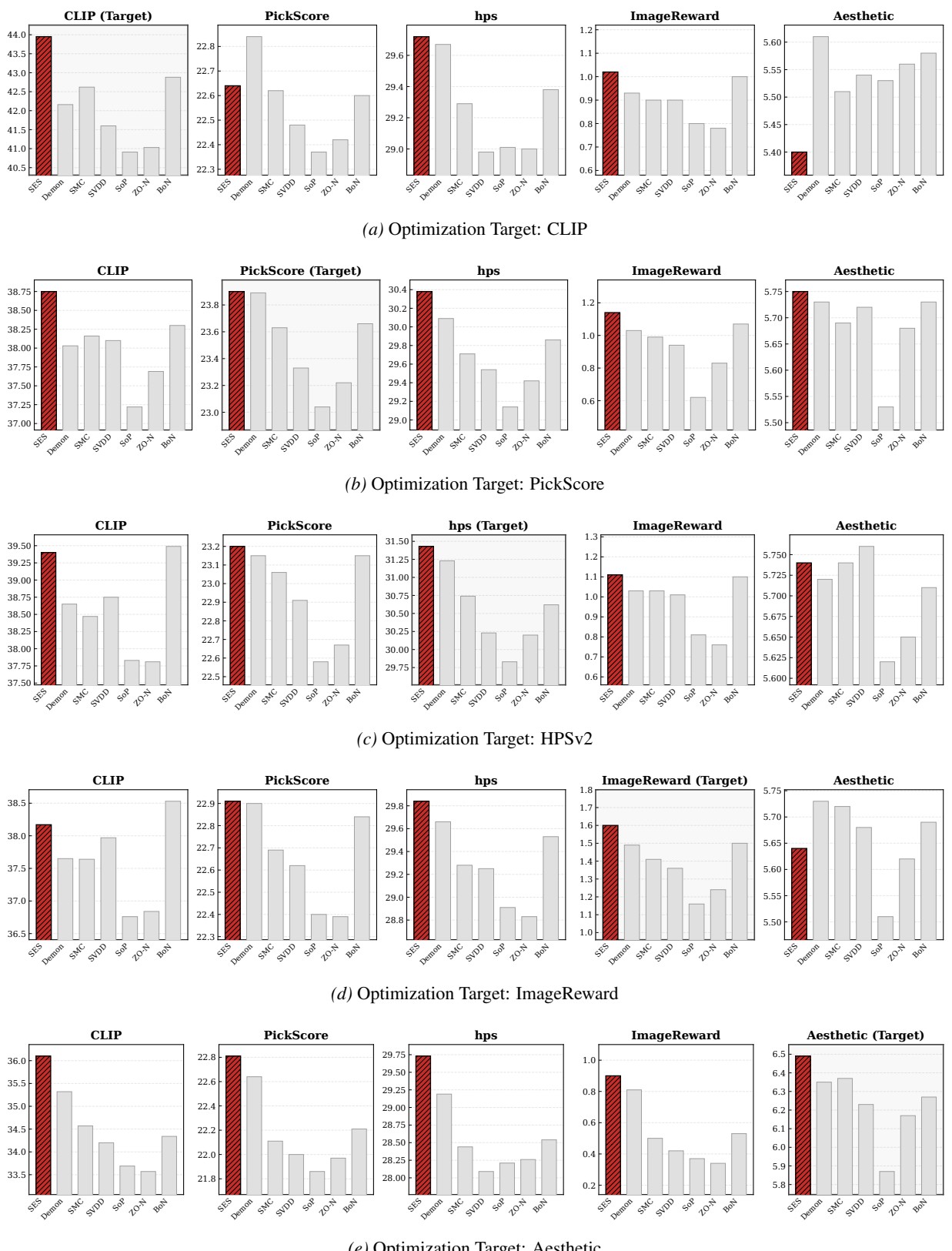

*Figure 12.* Cross-reward validation against reward hacking. Each row represents an experiment optimizing a specific target metric.

## F. Additional Qualitative Results

| Model | BoN | ZO-N | SoP | SMC | SVDD | Demon | SES |
|-------|-----|------|-----|-----|------|-------|-----|

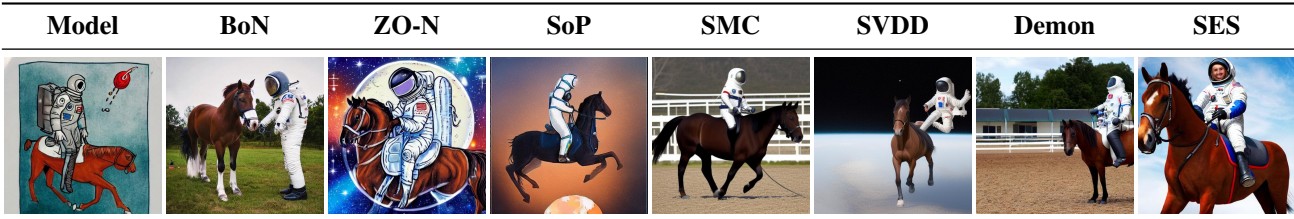

*SD v1.5 — Reward: HPSv2 — NRE = 200 — Prompt: A horse riding an astronaut.*

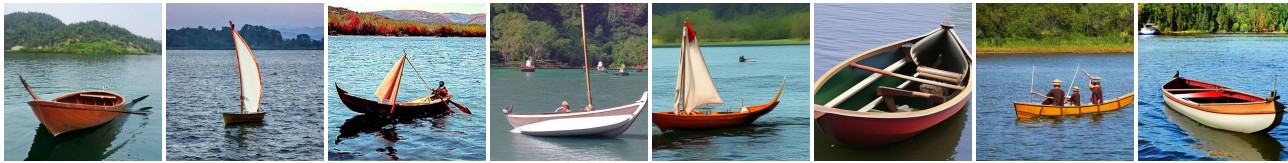

*SD v1.5 — Reward: HPSv2 — NRE = 200 — Prompt: A small vessel propelled on water by oars, sails, or an engine.*

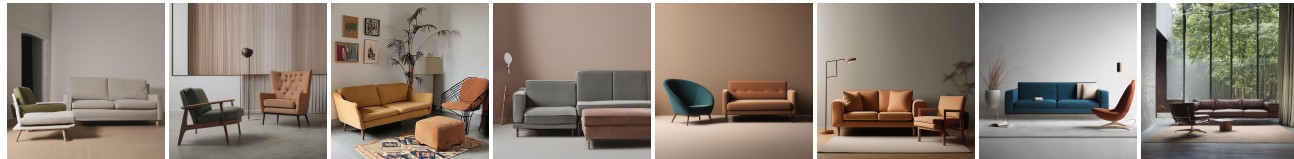

*SDXL — Reward: Aesthetic — NRE=1000 — Prompt: A couch on the right of a chair.*

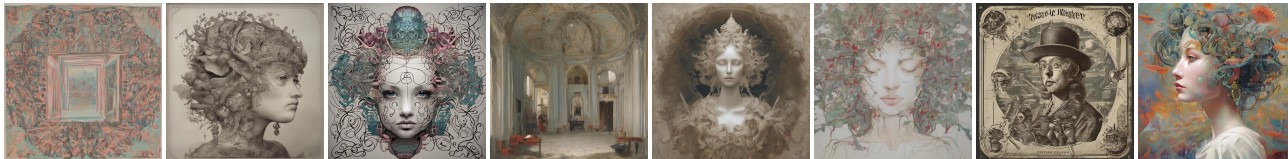

*SDXL — Reward: Aesthetic — NRE=1000 — Prompt: Artophagous.*

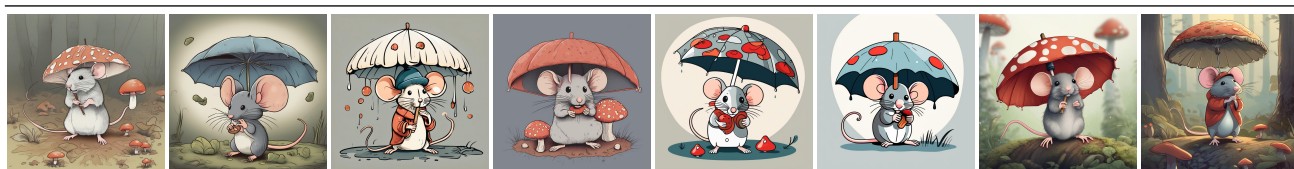

*SDXL — Reward: Aesthetic — NRE=1000 — Prompt: Illustration of a mouse using a mushroom as an umbrella.*

*Figure 13.* Additional qualitative comparisons on SD v1.5 and SDXL.

| Model | BoN | ZO-N | SoP | SES |
|-------|-----|------|-----|-----|

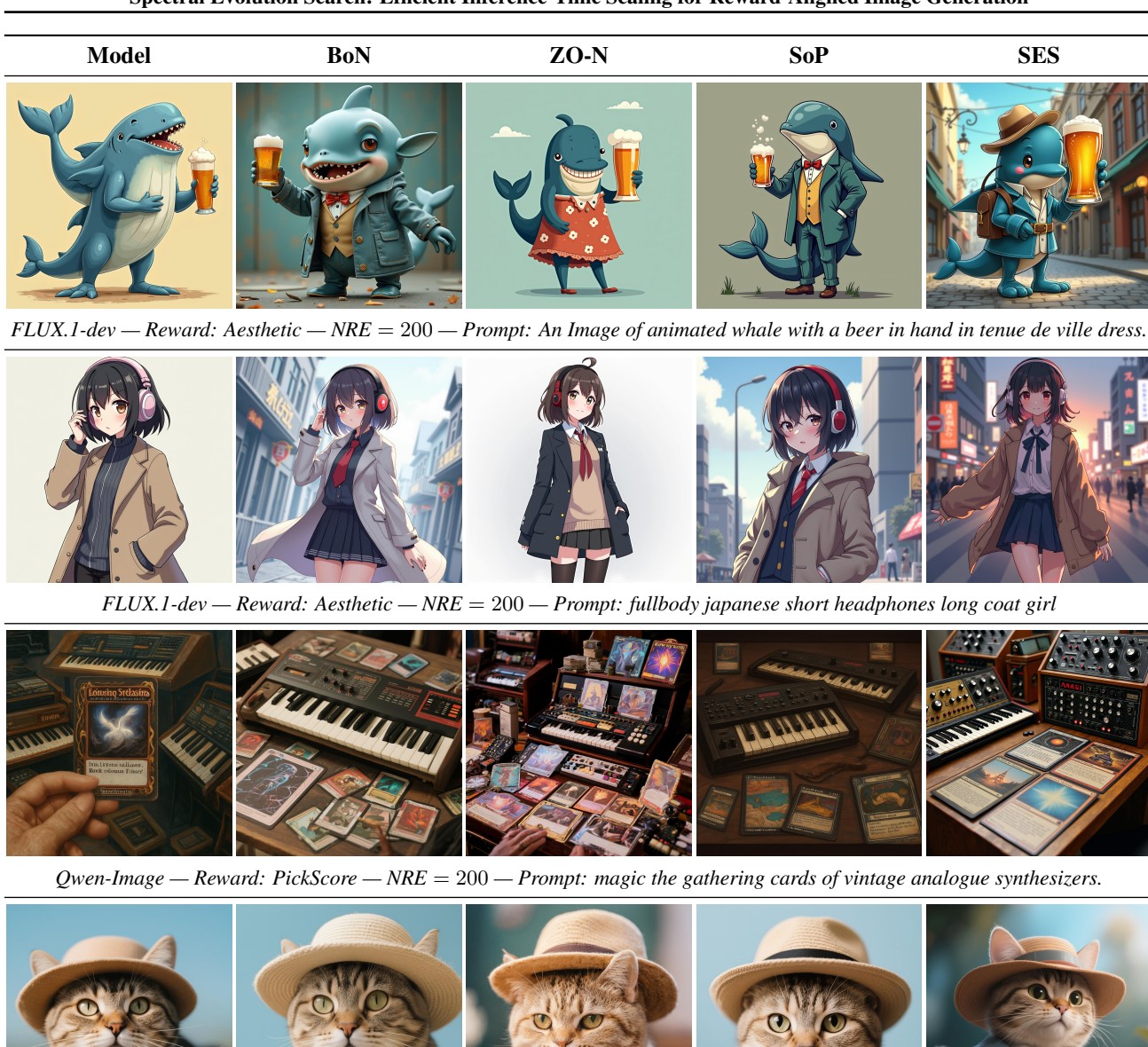

*FLUX.1-dev — Reward: Aesthetic — NRE = 200 — Prompt: An Image of animated whale with a beer in hand in tenue de ville dress.*

*FLUX.1-dev — Reward: Aesthetic — NRE = 200 — Prompt: fullbody japanese short headphones long coat girl*

*Qwen-Image — Reward: PickScore — NRE = 200 — Prompt: magic the gathering cards of vintage analogue synthesizers.*

*Qwen-Image — Reward: PickScore — NRE = 200 — Prompt: cat with a hat.*

*Figure 14.* Additional qualitative comparisons on FLUX.1-dev and Qwen-Image.

