# OpenReview forum: "Spectral Evolution Search: Efficient Inference-Time Scaling for Reward-Aligned Image Generation"
_ICML.cc/2026/Conference — ICML 2026 regular_

### Official Review · Reviewer_ie6Z · 2026-03-05

**Soundness:** 3
**Presentation:** 4
**Significance:** 4
**Originality:** 3
**Overall Recommendation:** 4
**Confidence:** 3

**Summary:**

This paper proposes **Spectral Evolution Search (SES)** to improve inference-time scaling in text-to-image generation. The key idea is to exploit spectral bias in generative models by restricting evolutionary search to the low-frequency subspace of the initial noise using wavelet decomposition, optimized via CEM. A theoretical Spectral Scaling Prediction is derived to motivate this design. Experiments on SDXL, FLUX.1-dev, and Qwen-Image show consistent gains over strong baselines under fixed compute budgets, with improved quality–diversity trade-offs and robustness to reward hacking.

**Compliance With Llm Reviewing Policy:**

Affirmed.

**Ethical Review Concerns:**

My concerns have been well addressed by the authors' rebuttal.

**Final Justification:**

My concerns have been well addressed in the authors' rebuttal. Given the careful and well-crafted formalization, which I consider the paper's main strength, I maintain my score in favor of (weak) acceptance.

**Key Questions For Authors:**

See weaknesses.

**Limitations:**

The theoretical analysis assumes approximate spectral separability and may overlook higher-order cross-frequency interactions. The method also depends on reward model quality and hyperparameter choices (e.g., decomposition level $J$ and smoothing factor $\gamma$). Although NRE is hardware-agnostic, actual runtime depends on reward evaluation cost, and overall scalability is not fully analyzed.

**Strengths And Weaknesses:**

Strengths:
1. The paper identifies a meaningful bottleneck in high-dimensional noise search and proposes a well-motivated low-frequency optimization strategy.
2. SES demonstrates consistent improvements across different model families and reward functions, and the proxy reward acceleration strategy is practically useful.
3. The theoretical analysis, connecting perturbation dynamics to power-law scaling behavior, adds scientific grounding beyond empirical observation.

Weaknesses:
1. While applying spectral bias to inference-time search is novel, prior work has documented similar spectral sensitivity in diffusion models. A clearer positioning relative to these studies would strengthen the contribution.
2. The theoretical derivation relies on simplifying assumptions (e.g., local linearization and approximate spectral decoupling), which may not strictly hold for modern transformer-based generators.
3. Performance is sensitive to the wavelet decomposition level, but no principled selection strategy is provided.

---

> ### Author Rebuttal · Authors · 2026-03-31
>
> Thank you for your careful reading and positive assessment of our paper. We respond to your concerns below.
>
> ---
> ## `W1` Relation to prior work on diffusion spectral bias
> **Short answer:** Prior work studies how frequencies are learned or denoised, whereas SES studies the spectral geometry of the `initial-noise optimization landscape`.
>
> Existing work on diffusion spectral bias generally falls into two distinct categories:
>
> - Training dynamics [1, 2]: Analyzing why models learn low frequencies first and how this relates to noise scheduling.
> - Inference dynamics [3,4]: Studying how different frequency components are recovered during denoising, and then using this to design frequency-aware guidance or acceleration methods.
>
> Our work shares with these studies the observation that diffusion models are frequency-anisotropic, but the object of study is different. We ask why different frequency directions in the initial noise have systematically different controllability and optimizability, and how this determines the effective degrees of freedom in reward-aligned black-box search. In this sense, SES is not simply an application of known spectral bias to a new task, but the study of a relatively underexplored object: the **spectral geometry of the initial-noise optimization landscape**.
>
> ---
> ## `W2 / L1` Applicability of the theory to modern architectures
> We acknowledge that our theory relies on frequency-domain decoupling, and therefore is not an exact description of the full dynamics of modern generators. However, SES does not rely on strict frequency independence. What it relies on is a more stable condition: perturbation gains are highly non-uniform across frequencies, and low-frequency directions have a larger average cumulative effect. As long as this holds, the effective degrees of freedom in the search space become spectrally anisotropic, and SES can exploit this structure to improve search efficiency. Our results on modern transformer-based architectures such as `FLUX` and `Qwen-Image` also show that, despite more complex cross-frequency interactions in real models, this dominant trend remains stable.
>
> ---
> ## `W3 / L2` How to choose the wavelet decomposition level
> **Short answer:** SES is not sharply sensitive to the decomposition level; in practice, a moderate intermediate level works well.
>
> We add further clarification on decomposition-level sensitivity. Under SDXL, with Aesthetic Score as the target reward and NRE = 200, the results for `L=3,4,5` are `6.38 / 6.50 / 6.47`, respectively. This indicates a stable plateau rather than dependence on a single precise level.
>
> The choice of decomposition level can be understood as a simple trade-off: if the level is too shallow (Table 2, L=2), too many high-dimensional degrees of freedom remain, which weakens the efficiency gain from dimensionality reduction; if the level is too deep (Table 2, L=6), the controllable subspace becomes overly compressed, which reduces the effective degrees of freedom. Therefore, a **reasonable intermediate level** is sufficient to balance search-space compactness and controllability.
>
> ---
> ## `L3` Runtime scalability
> To complement scaling, we add analyses based on real execution time. **Figure 8** already reports the `wall-clock time vs. target reward` on SDXL for SES and the baselines under both exact reward evaluation and proxy reward evaluation. The results show that, **at comparable wall-clock time, SES reaches a higher Target Reward Score**.
>
> We further compare the average runtime needed by each method to reach the same target score on 200 DrawBench prompts:
>
> - Under exact reward evaluation on SDXL, when reaching Aesthetic = 6.00 (base = 5.18), `SES takes 227.76s`, outperforming BoN (379.30s), ZO-N (493.62s), SoP (725.88s), SMC (271.93s), SVDD (314.04s), and Demon (263.29s).
> - Under proxy reward evaluation, the advantage of SES becomes even larger: `SES takes 85.62s`, compared with BoN (156.39s), ZO-N (163.42s), SoP (206.91s), SMC (132.01s), SVDD (179.17s), and Demon (101.25s).
> Therefore, the advantage of SES does not depend on NRE as the budget definition; it also holds under real wall-clock time.
>
> ---
>
> We hope these new wall-clock evaluations, ablation studies, and theoretical clarifications further strengthen your confidence in our contribution. If they do, we would be very grateful if you could consider supporting the paper more strongly. Thank you again for the helpful feedback.
>
> ---
> ## References
>
> [1] Wang, B., & Pehlevan, C. (2025). An analytical theory of spectral bias in the learning dynamics of diffusion models. arXiv preprint arXiv:2503.03206.
>
> [2] Falck et al. (2025). A fourier space perspective on diffusion models. arXiv preprint arXiv:2505.11278.
>
> [3] Sadat et al. (2025). Guidance in the frequency domain enables high-fidelity sampling at low cfg scales. arXiv preprint arXiv:2506.19713.
>
> [4] Liu et al. (2025). Freqca: Accelerating diffusion models via frequency-aware caching. arXiv preprint arXiv:2510.08669.

---

> > ### Author Rebuttal · Reviewer_ie6Z · 2026-04-01
> >
> > Thank you for the authors' detailed response. My concerns have been well addressed.

---

> > > ### Author Response · Authors · 2026-04-04
> > >
> > > Dear Reviewer ie6Z,
> > >
> > > Thank you for your time in reading our rebuttal and for your positive feedback. We are thrilled that your concerns have been well addressed. We truly appreciate your valuable feedback and support throughout the review process!

---

### Official Review · Reviewer_MMt1 · 2026-03-12

**Soundness:** 2
**Presentation:** 2
**Significance:** 2
**Originality:** 3
**Overall Recommendation:** 4
**Confidence:** 4

**Summary:**

This paper proposes an inference-time reward alignment method that optimizes the initial noise instead of modifying the generation trajectory. The main idea is to search only in a low-frequency wavelet subspace, based on the claim that image generators are much more sensitive to low-frequency perturbations. The paper supports this with both a spectral analysis and experiments on SDXL and FLUX.1-dev, showing improved reward optimization under fixed reward-evaluation budgets.

**Compliance With Llm Reviewing Policy:**

Affirmed.

**Final Justification:**

Most of my concerns are addressed.

**Key Questions For Authors:**

Please see weakness

**Limitations:**

Yes.

**Strengths And Weaknesses:**

**Strengths**

1. The paper tackles a practical problem, and the core idea is simple and easy to understand. Restricting the search to low-frequency noise is a clean design choice and seems broadly applicable.
2. The method is well-motivated and the paper is easy to read.

**Weakness**

1. The main concern is the choice of computational budget. The paper uses the number of reward evaluations (NRE) as the primary budget, but this does not reflect actual inference cost very well. For the proposed method, even a single reward evaluation is expensive, since it requires full generation together with multiple forward and backward passes through the denoising model. In contrast, many baselines optimize approximated rewards during sampling and typically require only one forward/backward update per denoising step, and BoN does not require backpropagation through the model at all. Because of these differences, matching methods only by NRE or NFE is not very convincing. I think the main results of Tab. 1 should be reported under matched wall-clock time, at least for SDXL and FLUX.1-dev.

2. Also, In Tab. 1, the authors only report the target reward scores. In inference-time reward alignment, reward hacking is a well-known issue: the target reward can improve while overall image quality degrades. For this reason, prior work typically reports the remaining reward metrics as held-out rewards. For example, when the target reward is Aesthetic Score, the held-out rewards would include metrics such as PickScore, ImageReward, and HPSv2, and similarly for other target rewards. I think those metrics should also be included for each experiment, ideally under matched wall-clock time.

3. Fig. 8 reports target reward vs. execution time for the few-step base model, which is useful. However, it would be much more convincing to also report the corresponding held-out rewards in this setting, too. That would help to verify that the improvement in target reward does not come at the expense of overall image quality.

4. The comparison is missing several very relevant baselines. There are several works that also optimize the initial noise for reward alignment while regularizing the search to remain close to the prior [1, 2, 3]. Since this paper is also an initial-noise optimization method, a direct comparison would be important to show that the gain really comes from the proposed search space and optimization scheme, rather than from the general idea of noise optimization itself. I think such comparisons should be included, ideally under matched wall-clock time.

[1] ReNO: Enhancing One-step Text-to-Image Models through Reward-based Noise Optimization, L. Eyring et al., NeurIPS 2024

[2] Tuning-Free Alignment of Diffusion Models with Direct Noise Optimization, Z. Tang et al., ICML 2024 Workshop

[3] Moment- and Power-Spectrum-Based Gaussianity Regularization for Text-to-Image Models, J. Hwang et al., NeurIPS 2025

---

> ### Author Rebuttal · Authors · 2026-03-31
>
> Thank you for your careful reading and concrete suggestions. We respond to your  concerns below.
>
> ## `W1` Fair budget comparison under wall-clock time
> We first clarify one key point: **SES is a fully gradient-free black-box search method**. It does not require backpropagation through either the generator or the reward model. Therefore, the cost of evaluating one candidate mainly consists of one forward generation and one reward evaluation, rather than the “multiple forward/backward passes through the denoising model” mentioned in the review.
>
> We use NRE because it provides a hardware-agnostic measure of the reward-query budget, especially when different methods use different proxy reward estimators at intermediate states. That said, we agree that NRE does not fully capture actual wall-clock latency. To address this, we add real runtime analyses. **Figure 8** already reports the `wall-clock time vs. target reward` on SDXL for SES and the baselines under both exact reward evaluation and proxy reward evaluation. The results show that, **at comparable wall-clock time, SES reaches a higher Target Reward Score**.
>
> We further compare the average runtime needed by each method to reach the same target score on 200 DrawBench prompts:
>
> - Under exact reward evaluation on SDXL, when reaching Aesthetic = 6.00 (base = 5.18), `SES takes 227.76s`, outperforming BoN (379.30s), ZO-N (493.62s), SoP (725.88s), SMC (271.93s), SVDD (314.04s), and Demon (263.29s).
>
> - Under proxy reward evaluation, the advantage of SES becomes even larger: `SES takes 85.62s`, compared with BoN (156.39s), ZO-N (163.42s), SoP (206.91s), SMC (132.01s), SVDD (179.17s), and Demon (101.25s).
>
> Therefore, the advantage of SES does not depend on NRE as the budget definition; it also holds under real wall-clock time.
>
> ---
> ## `W2 & W3` Reward hacking and held-out rewards
> **Short answer:** We already perform cross-reward evaluation in Appendix E.2, and SES remains strong not only on the target reward but also on held-out reward models.
>
> We have already conducted a dedicated `cross-reward evaluation in Appendix E.2`. When optimizing one target reward, we simultaneously monitor performance on other held-out reward models that are not used for optimization. The results show that SES not only remains best on the target reward, but also usually outperforms all baselines on most held-out metrics. This suggests that the gains of SES do not come from adversarially exploiting a single reward model, but from broader improvements in image quality.
>
> For `W3`, we report the corresponding held-out metrics. When the target is Aesthetic, SES reaches Aesthetic = 6.19 while also achieving PickScore = 22.64, HPS = 29.47, ImageReward = 1.08, and CLIP = 38.25. This confirms that the increase in target reward does not come at the expense of overall perceptual quality.
>
> ---
> ## `W4` Compare with initial-noise optimization baselines
> For `DNO` (the formal ICML 2025 version of your ref `[2]`), we already discussed this work in `Appendix E.1`. We observed that it exhibits clear reward hacking after just ~20 iterations. Thus, we could not include it for a fair quantitative comparison.
>
> For `ReNO` (your ref `[1]`) and `Hwang et al.` (your ref `[1]`), we conduct direct comparisons. Since these works are specifically designed for distilled, few-step models, we evaluated them on `SDXL-Turbo (1-step)` and `FLUX.1-schnell (4-step)` under exactly matched budgets (NRE=100, Target: PickScore)  and also include the corresponding wall-clock time.
>
> |Model|Method|Pick|HPS|ImgRew.|Aes.|CLIP|Time|
> |:-|:-|:-:|:-:|:-:|:-:|:-:|:-:|
> |SDXL-Turbo|Base|22.29|27.95|-0.86|5.56|34.84|0.26|
> ||ReNO|23.62|28.81|1.01|5.61|37.16|35.15|
> ||Hwang et al.|23.60|28.98|1.07|5.62|37.25|34.91|
> ||**SES**|`23.77`|**29.24**|**1.14**|**5.65**|**37.79**|`26.45`|
> |FLUX.1-schnell|Base|22.68|29.64|-0.59|5.57|36.27|1.12|
> ||ReNO|23.84|29.51|0.91|5.07|**38.53**|113.37|
> ||Hwang et al.|23.90|29.67|0.97|5.41|38.42|113.14|
> ||**SES**|`23.99`|**29.89**|**1.28**|**5.77**|38.15|`112.65`|
>
> As shown, SES achieves the best target reward, remains strong on held-out metrics, and is also better in wall-clock time against these initial-noise optimization baselines.
>
> Another important difference is that ReNO and Hwang et al. are mainly designed for distilled fast-generation models and rely on gradient ascent to update the initial noise, which requires a differentiable reward. In contrast, SES is a fully gradient-free black-box method. Its computational overhead beyond generation and reward evaluation is very small. Moreover, it applies **more broadly** to generative processes parameterized by initial noise, and it can be used directly with non-differentiable reward signals.
>
> ---
>
> We hope these additional runtime analyses, held-out reward evaluations, and new baseline comparisons address your concerns. If they do, we would be very grateful if you could reconsider your assessment and support the paper more. Thanks again!

---

> > ### Author Rebuttal · Reviewer_MMt1 · 2026-04-03
> >
> > Thank you for your detailed response.
> >
> > Most of my concerns are addressed, thus I'll raise my score.

---

> > > ### Author Response · Authors · 2026-04-04
> > >
> > > Dear Reviewer MMt1,
> > >
> > > Thank you very much for reading our rebuttal and for raising your score! We are encouraged that our new experiments and clarifications successfully addressed your core concerns. Your constructive suggestions have strengthened our paper. Thank you again for your time and support!

---

### Official Review · Reviewer_4umM · 2026-03-13

**Soundness:** 3
**Presentation:** 3
**Significance:** 3
**Originality:** 3
**Overall Recommendation:** 4
**Confidence:** 3

**Summary:**

This paper investigates the issue of inference time extension in reward-aligned image generation and proposes the "Spectral Evolution Search" (SES) framework without gradient optimization. The core concept analyzed in the article is that the low-frequency perturbation of the initial noise has a much stronger impact on the final generated image than the high-frequency perturbation. Based on this, this method decomposes the initial noise based on wavelets, freezes the high-frequency components, and conducts cross-entropy evolutionary search only in the low-frequency subspace to improve the reward score. Overall, one related issue evaluated in this paper is how to effectively convert additional inference computations into better reward-aligned generation effects without modifying the model parameters.

**Compliance With Llm Reviewing Policy:**

Affirmed.

**Final Justification:**

The authors successfully addressed my initial concerns during the rebuttal. I keep my score and recommend a Weak Accept

**Key Questions For Authors:**

1. SES restricts optimization to the low-frequency subspace based on the spectral bias observation. However, it is unclear whether this restriction might limit the model's ability to capture fine-grained visual details. Could the authors provide additional analysis or experiments on prompts that require high-frequency details?

2. How sensitive is SES to the choice of wavelet transform or decomposition level? If different frequency decompositions are used, would the performance remain stable?

3. Recent studies have explored optimizing other potential variables, such as the embeddings used in the classifier-free approach, rather than the initial noise method. Have the authors considered comparing SES with these methods, or analyzing whether there are similar spectral characteristics in these potential spaces?

**Limitations:**

Yes

**Strengths And Weaknesses:**

## Strengths

1. This paper points out an important limitation of the existing reasoning time search methods. Optimizing in the complete high-dimensional noise space leads to a large amount of redundancy and poor expansion efficiency.

2. Restricting optimization to the low-frequency subspace via wavelet decomposition is intuitive and easy to implement. The approach is plug-and-play and can be integrated with various diffusion models and reward functions.

3. The spectral scaling prediction derived from perturbation propagation offers a theoretical explanation for why low-frequency perturbations dominate generation control, which helps justify the design of SES.

## Weaknesses

1. The core framework of this paper still adheres to the initial noise optimization paradigm, aiming to enhance reward alignment by optimizing the initial noise of the diffusion model before sampling. In recent years, several studies have systematically explored this direction. For instance, Direct Noise Optimization proposes to directly optimize the noise variables during the sampling process to improve the reward alignment effect. Compared with these methods, the main contribution of this paper lies in reducing the search space dimension through frequency domain decomposition, while the overall optimization framework is relatively similar.

2. The core assumption of SES is that low-frequency noise has a stronger influence on the generation results, so the search is restricted to the low-frequency subspace. However, in some scenarios that require fine-grained texture or local structure control, high-frequency perturbations may still play an important role. The current paper does not analyze whether this low-frequency limitation will cause problems such as insufficient generation details or decreased diversity in certain tasks. If more analysis or visualization experiments for different types of prompts can be added, or if further discussions on the role of high-frequency components can be provided, it will make the applicability of the method clearer.

3. The spectral scaling prediction proposed in the paper provides an interesting explanation for the method, but the derivation process relies on some strong assumptions, such as the approximation of the Jacobian of the denoising network and the assumption of frequency-domain decoupling. Currently, the paper mainly supports these conclusions through empirical results, but the relationship between theory and the actual model behavior is still not completely clear.

---

> ### Author Rebuttal · Authors · 2026-03-31
>
> Thank you for your careful reading and positive feedback. We address each Weakness (W) and Question (Q) below.
>
> ---
> ## `W1` SES vs. prior initial-noise optimization methods
> Although SES belongs to the initial-noise optimization paradigm, it addresses two core bottlenecks of this paradigm.
>
> - For **gradient-based methods** (e.g., Direct Noise Optimization), high-frequency directions are more prone to adversarial perturbations in practice, which can increase the risk of reward hacking. By freezing the high-frequency components, SES constrains the search to more safe directions.
>
> - For **gradient-free search methods**, isotropic exploration in the full space suffers from the curse of dimensionality. SES restricts the search to the low-frequency subspace, so that more of the budget is spent on the effective control directions.
>
> Therefore, the key contribution of SES is not merely to optimize the initial noise, but to `redesign the search space` itself by exploiting its spectrally non-uniform controllability structure.
>
> ---
> ## `W2 / Q1` Does SES hurt high-frequency details
> **Short answer:** We do not observe a systematic loss of high-frequency details.
>
> Our claim is not that high-frequency details are unimportant. Rather, in initial-noise optimization, high-frequency directions tend to be inefficient dimensions for search, because their controllable effect on the final reward is substantially weaker than that of low-frequency directions. SES only restricts which directions of the initial noise are optimized at test time; it does **not** modify the pretrained generator, the denoising network, or the sampling process. Therefore, fine-grained textures and local structures are still naturally recovered during generation.
>
> We add prompt-level visual analysis for prompts emphasizing fine-grained textures and local structures. SES shows no systematic loss of high-frequency details compared with standard generation, while still improving the target reward. Visualizations are provided at the anonymous link: https://anonymous.4open.science/r/SES-rebuttal-513D/README.md.
>
> ---
> ## `W3` Relation between theory and real model behavior
> The Spectral Scaling Prediction relies on local Jacobian approximation together with approximate frequency-domain decoupling, and is therefore not intended to serve as  a full dynamical model of transformers-based diffusion models. Its goal is more specific: to explain a stable dominant phenomenon, namely, that the influence of an initial perturbation on the final output decays systematically with frequency.
>
> In this sense, the theory should be interpreted as a **structural explanation** of the scaling law, rather than an exact predictor for every layer and every frequency band. Even if real models exhibit higher-order nonlinearities and cross-frequency interactions, the search structure exploited by SES still holds as long as the dominant sensitivity decays with frequency. This is directly supported by the frequency-sensitivity trends measured from real models in Figures 1 and 3.
>
> ---
> ## `Q2` Sensitivity to wavelet type and decomposition level
> **Short answer**: SES is robust to both the decomposition method and the decomposition level.
>
>
> For the **decomposition level**, under SDXL with Aesthetic Score as the target reward and NRE = 200, the results for `L=3,4,5` are `6.38 / 6.50 / 6.47`, respectively. This indicates a stable plateau over a reasonable range, rather than dependence on a single precise level.
>
> For the **transform choice**, the mainstream `db / sym / coif` wavelet families all yield stable results. In addition, replacing wavelets with DCT leads to very similar performance. This suggests that SES does not rely on an incidental property of one particular wavelet, but on the more general existence of a dominant low-frequency subspace.
>
> ---
> ## `Q3` Extension to other optimizable variables
> We have not yet conducted systematic experiments on other latent variables, so we do not claim direct applicability there. We focus on the initial noise because it resides on an image-like spatial grid, where Fourier/wavelet frequency provides a natural and interpretable coordinate system.
>
> In contrast, semantic latent variables such as classifier-free guidance embeddings do not have a natural notion of spatial frequency, so the low-/high-frequency decomposition used here does not transfer directly. More broadly, such spaces may still exhibit anisotropy in controllability, but it is more likely to appear as principal directions, low-rank subspaces, or other structured bases rather than spectrum in the spatial-frequency sense. What may transfer is the broader principle of **structured subspace search**, not the current low-frequency wavelet parameterization itself.
>
> ---
>
> We hope these clarifications address your concerns. If so, we would be very grateful if you could consider supporting the paper more strongly. Thank you again for the helpful feedback.

---

> > ### Author Rebuttal · Reviewer_4umM · 2026-04-02
> >
> > Thanks for the detailed rebuttal, which successfully addresses my initial concerns.

---

> > > ### Author Response · Authors · 2026-04-04
> > >
> > > Dear Reviewer 4umM,
> > >
> > > Thank you for your time in reading our rebuttal and for your positive feedback. We are glad that our response has successfully addressed your concerns. Thank you again for your constructive guidance and support!

---

### Official Review · Reviewer_UqEk · 2026-03-13

**Soundness:** 3
**Presentation:** 3
**Significance:** 2
**Originality:** 3
**Overall Recommendation:** 4
**Confidence:** 4

**Summary:**

This paper studies the test/inference-time scaling of flow and diffusion-based visual generative models by optimizing the initial noise rather than the denoising trajectory. The main idea behind the paper is based on spectral bias, which specifically refers to the observation that low-frequency perturbations of the initial noise have much larger downstream impact on the final image than high-frequency perturbations. The plug-and-play framework developed in this paper is called Spectral Evolution Search (SES) by restricting the search space to a low-frequency subspace, which is constructed via a decomposition based on discrete wavelet transform. Specifically, SES adopts the cross-entropy method (CEM), which is a gradient-free evolutionary strategy following the "sample-evaluate-update" loop, to search for the optimal noise within the subspace. Theoretically, the authors provide an argument to justify how the influence of perturbations scales inversely with respect to the frequency via the perturbation dynamics and approximation within the frequency domain. Empirically, SES is evaluated on multiple text-to-image generative models like SDXL and FLUX with respect to several reward objectives such as CLIP, PickScore, HPS and ImageReward. The reported results show that SES consistently improves reward alignment under fixed computational budgets and exhibits stronger scaling behavior than several sampling and search-based baselines.

**Compliance With Llm Reviewing Policy:**

Affirmed.

**Final Justification:**

The reviewer has checked the authors' rebuttals and founded that almost all of the reviewers' concerns have been resolved. Specifically, the reviewer is satisfied with the authors' clarification on the novelty of this work, especially the comparison between SES and inference-time scaling/guidance methods. Therefore, the reviewer is willing to increase the score from 3 to 4.

**Key Questions For Authors:**

Please refer to the "Weaknesses" subsection above.

**Limitations:**

Regarding limitations, the authors may refer to the "Weaknesses" and "Questions" sections above. Since this is a paper mainly about inference-time alignment and generative AI (diffusion models), the reviewer doesn't think there is any negative societal impact of the work.

Overall, I think this is a solid and practically interesting submission with interesting theoretical results and good empirical evidence, but I am somewhat hesitant about whether the methodological novelty is strong enough for ICML in its current form. The authors are strongly encouraged to carefully incorporate all suggestions, discuss all issues in detail, and include all missing references listed below.

References:

[1] Wang, Binxu, and Cengiz Pehlevan. "An analytical theory of spectral bias in the learning dynamics of diffusion models." arXiv preprint arXiv:2503.03206 (2025).

[2] Zhang, Xiangcheng, Haowei Lin, Haotian Ye, James Zou, Jianzhu Ma, Yitao Liang, and Yilun Du. "Inference-time scaling of diffusion models through classical search." arXiv preprint arXiv:2505.23614 (2025).

[3] Ramesh, Vignav, and Morteza Mardani. "Test-time scaling of diffusion models via noise trajectory search." arXiv preprint arXiv:2506.03164 (2025).

[4] Tang, Sophia, Yuchen Zhu, Molei Tao, and Pranam Chatterjee. "Tr2-d2: Tree search guided trajectory-aware fine-tuning for discrete diffusion." arXiv preprint arXiv:2509.25171 (2025).

[5] Uehara, Masatoshi, Yulai Zhao, Chenyu Wang, Xiner Li, Aviv Regev, Sergey Levine, and Tommaso Biancalani. "Inference-time alignment in diffusion models with reward-guided generation: Tutorial and review." arXiv preprint arXiv:2501.09685 (2025).

[6] Chen, Haoxuan, Yinuo Ren, Martin Renqiang Min, Lexing Ying, and Zachary Izzo. "Solving inverse problems via diffusion-based priors: An approximation-free ensemble sampling approach." arXiv preprint arXiv:2506.03979 (2025).

[7] Skreta, Marta, Tara Akhound-Sadegh, Viktor Ohanesian, Roberto Bondesan, Alán Aspuru-Guzik, Arnaud Doucet, Rob Brekelmans, Alexander Tong, and Kirill Neklyudov. "Feynman-kac correctors in diffusion: Annealing, guidance, and product of experts." arXiv preprint arXiv:2503.02819 (2025).

[8] Kim, Yeongmin, Donghyeok Shin, Byeonghu Na, Minsang Park, Richard Lee Kim, and Il-Chul Moon. "Lookahead Sample Reward Guidance for Test-Time Scaling of Diffusion Models." arXiv preprint arXiv:2602.03211 (2026).

[9] Singhal, Raghav, Zachary Horvitz, Ryan Teehan, Mengye Ren, Zhou Yu, Kathleen McKeown, and Rajesh Ranganath. "A general framework for inference-time scaling and steering of diffusion models." arXiv preprint arXiv:2501.06848 (2025).

[10] Feng, Shengyu, Xiang Kong, Shuang Ma, Aonan Zhang, Dong Yin, Chong Wang, Ruoming Pang, and Yiming Yang. "Step-by-step reasoning for math problems via twisted sequential monte carlo." arXiv preprint arXiv:2410.01920 (2024).

[11] Zhao, Stephen, Rob Brekelmans, Alireza Makhzani, and Roger Grosse. "Probabilistic inference in language models via twisted sequential monte carlo." arXiv preprint arXiv:2404.17546 (2024).

[12] Lee, Cheuk Kit, Paul Jeha, Jes Frellsen, Pietro Lio, Michael Samuel Albergo, and Francisco Vargas. "Debiasing guidance for discrete diffusion with sequential monte carlo." arXiv preprint arXiv:2502.06079 (2025).

[13] Ou, Zijing, Chinmay Pani, and Yingzhen Li. "Inference-Time Scaling of Discrete Diffusion Models via Importance Weighting and Optimal Proposal Design." arXiv preprint arXiv:2505.22524 (2025).

**Strengths And Weaknesses:**

Strengths: The reviewer believes that this paper studies an important and popular problem (inference-time alignment for vision generative models). Also, the intuition that restricting the search space to a small subset based on wavelet decomposition is appealing. To the best of the reviewer's knowledge, such idea probably hasn't been explored in previous work on the inference-time scaling of generative models. Also, the empirical results, which span across multiple architectures and several reward models, appear to be very solid.

Weaknesses: The reviewer is first a bit concerned about the strength of the theory. Though the theory supports the intuition behind SES, it does not rigorously prove that the chosen low-frequency wavelet subspace is near-optimal for search in nonlinear models. Also, would it be possible for the authors to comment on how the theoretical results relate to existing studies on the spectral bias of diffusion models, such as [1]? It seems that the reference is currently missing.

Another concern that the reviewer has is that currently the manuscript has missed a lot of references and related work. For instance, regarding the class of sampling-based methods for reward alignment, there has been lots of work developing SMC-based approaches for priors encoded by either discrete or continuous diffusion models. An incomplete list of missing references include [5,6,7,8,9] (continuous diffusion model) and [10,11,12,13] (discrete diffusion model and LLMs). Another instance is search-based methods for diffusion model alignment, where examples of missing references include [2,3,4].

---

> ### Author Rebuttal · Authors · 2026-03-31
>
> Thank you for your careful reading and thoughtful feedback. We address your comments below.
>
> ---
> ## `W1` Theoretical strength
> We agree that the current analysis does not rigorously prove that the low-frequency wavelet subspace is “near-optimal” for arbitrary nonlinear generative models. However, this is not the claim we intend to make. Instead, our theory supports a more precise statement that is better aligned with the inference-time search setting: under a fixed search budget, different frequency directions in the initial noise have systematically different levels of control over the final generation, and low-frequency directions are more search-efficient.
>
> In this sense,  our work characterizes a **budget-constrained controllability advantage** rather than asserting global optimality. Our theory is intended to reveal the anisotropic structure of the initial-noise optimization landscape and to explain why isotropic search wastes substantial budget in directions with negligible influence. This insight is consistently supported by empirical evidence: low-frequency subspaces outperform the full space, high-frequency subspaces, and random subspaces of matched dimensionality (Table 2).
>
> ---
>
> ## `W2` Relation to prior work on diffusion spectral bias
> **Short answer:** Prior work studies how frequencies are learned or denoised, whereas SES studies the spectral geometry of the `initial-noise optimization landscape`.
>
> Existing work on diffusion spectral bias generally falls into two distinct categories:
> - Training dynamics [1, 2]: Analyzing why models learn low frequencies first and how this relates to noise scheduling.
> - Inference dynamics [3,4]: Studying how different frequency components are recovered during denoising, and then using this to design frequency-aware guidance or acceleration methods.
>
> While our work shares with these studies the empirical observation that diffusion models exhibit strong frequency anisotropy, the object of study isdifferent. Prior work examines frequency learning or recovery along the sampling trajectory. In contrast, SES focuses on **the spectral structure of the initial-noise search space itself**-specifically, why different frequency directions have systematically different controllability and optimizability, and how this determines the effective degrees of freedom in black-box search.
>
> Therefore, SES should not be viewed as a straightforward application of known spectral bias to a new task. Rather, it identifies and exploits a previously underexplored aspect: the **spectral geometry of the initial-noise optimization landscape**.
>
> ---
> ## `W3` Position of SES within inference-time scaling methods
> Inference-time scaling methods can be broadly categorized into three groups. **(1) Resampling-based methods** operate on intermediate trajectory states through resampling, particle correction, or proposal design (e.g., the works you listed in [5–13]). **(2) Search-based methods** explicitly search over the generation trajectory, such as classical search or tree search (e.g., the works you listed in [2–4]). **(3) Initial-noise optimization methods** directly optimize the initial noise of a fixed generator, either with gradient-based updates or with gradient-free search.
>
> SES belongs to the third category, specifically in the gradient-free black-box setting. Our key contribution is to identify a fundamental bottleneck in this paradigm: the high-dimensional initial noise is typically parameterized isotropically, which wastes substantial search budget on directions with little influence on the final output. SES addresses this limitation by exploiting spectral anisotropy to restructure the search space itself, thereby improving search efficiency.
>
> From this perspective, the SMC-based and trajectory-search methods are complementary rather than competing with our work. These methods focus on how to reweight or search along trajectories, whereas SES addresses a distinct and orthogonal question: **how the initial noise should be optimized**.
>
> ---
>
> We hope these additional related work analyses and theoretical clarifications address your concerns. If they do, we would be very grateful if you could reconsider your assessment and support the paper more. Thanks again!
>
> ---
> ## References
>
> [1] Wang, B., & Pehlevan, C. (2025). An analytical theory of spectral bias in the learning dynamics of diffusion models. arXiv preprint arXiv:2503.03206.
>
> [2] Falck, F., Pandeva, T., Zahirnia, K., Lawrence, R., Turner, R., Meeds, E., ... & Karmalkar, S. (2025). A fourier space perspective on diffusion models. arXiv preprint arXiv:2505.11278.
>
> [3] Sadat, S., Vontobel, T., Salehi, F., & Weber, R. M. (2025). Guidance in the frequency domain enables high-fidelity sampling at low cfg scales. arXiv preprint arXiv:2506.19713.
>
> [4] Liu, J., Cai, P., Zhou, Q., Lin, Y., Kong, D., Huang, B., ... & Zhang, L. (2025). Freqca: Accelerating diffusion models via frequency-aware caching. arXiv preprint arXiv:2510.08669.

---

### Decision · Program_Chairs · 2026-04-30

**Decision:**

Accept (regular)

**Comment:**

This paper proposes a plug-and-play initial noise optimization framework that exploits the spectral bias of diffusion models by restricting evolutionary search to a low-frequency wavelet subspace. After the rebuttal, all reviewers gave positive scores, and all concerns were marked as fully or mostly resolved. Reviewers appreciated the well-motivated and theoretically grounded core idea (low-frequency subspace search), as well as the strong empirical results across multiple models and reward functions.

For the final version, it is strongly recommended to incorporate the missing related-work references (SMC-based and search-based methods), include the wall-clock time comparisons and held-out reward evaluations from the rebuttal in the main paper, and clarify the theoretical assumptions (local linearization and approximate spectral decoupling) and their limitations for modern transformer-based architectures.